# Topological Invariance and Breakdown in Learning

## Abstract

We prove that for a broad class of permutation-equivariant learning rules (including SGD, Adam, and others), the training process induces a bi-Lipschitz mapping of neurons and preserves key topological properties of the neuron distribution. This result reveals a qualitative difference between small and large learning rates. Below a critical topological threshold $\eta^*$, the training is constrained to preserve the topological structure of the neurons, whereas above $\eta^*$ the process allows topological simplification, making the neuron manifold progressively coarser and reducing the model's expressivity. An important feature of our theory is that it's independent of specific architectures or loss functions, enabling universal applications of topological methods to the study of deep learning.

## 1 Introduction

Deep learning has emerged as an extraordinarily powerful tool, yet due to its complexity and inherent nonlinearity, our understanding of its inner mechanisms remains limited. There is a strong practical motivation for studying learning dynamics: a unified understanding of learning dynamics could inform the design of new regularization techniques, learning-rate schedule algorithms and other training strategies, thereby reducing the reliance on extensive hyperparameter tuning and facilitating the development of more efficient models (Sutskever et al., 2013; Gotmare et al., 2018; Liu et al., 2019; Kalra & Barkeshli, 2024). More recently, numerous empirical works have described the universal aspects of learning dynamics (Zhou et al., 2025; Cohen et al., 2021; Gur-Ari et al., 2018), yet a unified theoretical framework is still lacking.

A primary difficulty in analyzing the learning dynamics of neural networks lies in their extremely high dimensionality across diverse architectural details. Modern neural networks, such as GPT-4, have more than $10^{12}$ parameters, inducing such complicated dynamics that conventional tools and theories of dynamical systems struggle to apply. Lessons from natural science and many fields of mathematics suggest two primary approaches (Noether, 1918; Mumford et al., 1994): (1) study what the high-dimensional object is invariant to, and (2) decompose it into simpler parts. The first approach directly reduces the dimensionality of a problem, while the second allows us to view it as a composition of low-dimensional objects. Our theory, presented in this paper, aims to offer a crucial link between the two perspectives and show that due to a universal property, the permutation invariance (or equivariance) of the model (or learning algorithm), almost any neural network can be naturally decomposed into a system of interacting "neurons" with much smaller dimensions.

Specifically, under standard regularity conditions, we show that:
1. The permutation equivariance of common learning algorithms imposes strong topological constraints on the learning dynamics;[1]
2. With a small learning rate $\eta$, the learning algorithm induces a bi-Lipschitz mapping between neurons at different time steps, thereby preserving the topological structure of the set formed by the neurons.
3. With a large $\eta$, this topological invariance breaks down: the learning algorithm descends to a continuous surjection, thereby inducing a simplification process during training.

The core contribution of this paper can be summarized as the theoretical establishment of a critical point of the topological phase transition (hereinafter referred to as the **topological critical point**) for learning processes. Figure 1 illustrates the theory.

---

[1]The word "topology" is sometimes used to refer to the model architecture. In our work, it always means the mathematical topology of sets.

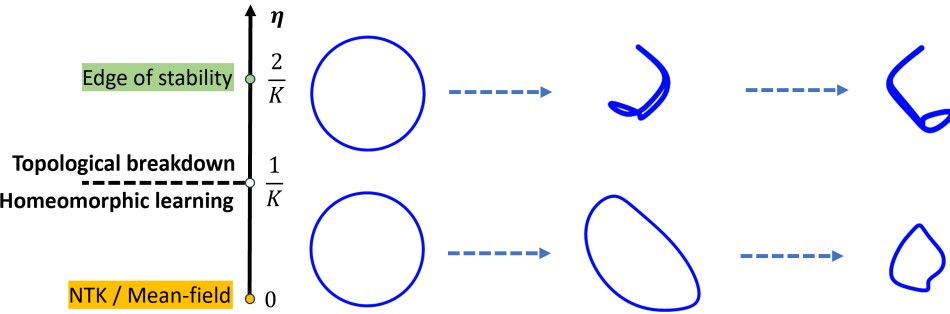

Figure 1: At a small learning, common learning algorithms induce a homeomorphic transformation of the neuron distribution (blue shapes in Figure), a mechanism underlying common theories including the NTK / lazy regime (Jacot et al., 2018; Chizat et al., 2018) and the mean-field / feature-learning regime (Yang & Hu, 2020). In contrast, most neural networks in real training scenarios are known to move towards the "edge of stability," where the discrete-time updates are no longer stable at any first-order stationary point. From the perspective of topology, what separates these two regimes is the **topology invariance** in the first regime, where the learning process is strongly constrained to preserve any topological properties, and the **topological breakdown** in the second regime, where the learning ceases to preserve topology and acts as a simplifier that merges neurons and makes the model more and more constrained in capacity.

A key feature of our theory is that it relies only on several widely satisfied properties of the learning algorithm and is therefore universal across architectures and optimizers. The system-independence of our result allows us to establish a key conceptual principle: *any permutation-equivariant dynamics induces a topology between its components, and this topology is preserved at a small step size and reduced at a large step size*. This universality lends it good potential to serve as a foundation for future theories. Moreover, our framework is firmly grounded in mathematical topology and can be further developed using tools thereof (Milnor & Weaver, 1997). The close connection between topology and theoretical physics also opens the door for relevant concepts from physics to be applied here (Qi & Zhang, 2011).

This paper is organized as follows. Section 2 reviews the background. Section 3 introduces the problems setting. Section 4 presents our main theory. Section 5 applies the theory to common training algorithms. Section 6 presents empirical results. Finally, Section 7 discusses further implications of our theory. All proofs of the theoretical results are deferred to the appendix.

## 2 BACKGROUND

**Permutation Symmetry.** Permutation symmetry refers to the invariance of a function's output under permutations of its inputs. This property is pervasive in neural networks and has been widely used to analyze their loss landscapes (Entezari et al., 2021; Brea et al., 2019; Ziyin, 2024). For example, any neural network component (such as a layer) that has the following structure:

$$f(\boldsymbol{x}; \boldsymbol{W}_1, \boldsymbol{W}_2) = \boldsymbol{W}_2 \sigma (\boldsymbol{W}_1 \boldsymbol{x}), \tag{1}$$

where $\boldsymbol{W}_1, \boldsymbol{W}_2$ are learnable parameter matrices, $\boldsymbol{x}$ is the input vector and $\sigma$ is a scalar activation function (applied element-wisely), possesses permutation symmetry, as

$$f(\boldsymbol{x}; \boldsymbol{W}_1, \boldsymbol{W}_2) = \left(\boldsymbol{P}\boldsymbol{W}_2^\top\right)^\top \sigma \left((\boldsymbol{P}\boldsymbol{W}_1)\boldsymbol{x}\right) = f(\boldsymbol{x}; \boldsymbol{P}\boldsymbol{W}_1, \boldsymbol{W}_2\boldsymbol{P}^\top) \tag{2}$$

for any permutation matrix $\boldsymbol{P}$. If we pair the $i$-th row of $\boldsymbol{W}_2$ with the $i$-th column of $\boldsymbol{W}_1$ together as a unit (which together form a "neuron" in our theory), then the symmetry can be understood as that, the model remains unchanged under exchanging two neurons $(\boldsymbol{w}_{1,i}, \boldsymbol{w}_{2,i}) \leftrightarrow (\boldsymbol{w}_{1,j}, \boldsymbol{w}_{2,j})$.

Structures in the form of eq. (1) are quite common in all types of neural networks, including convolutional layers, feed-forward layers, and the QK transformation in the self-attention layers in transformers. Specifically, the QK transformation in transformers can be represented as

$$f(\boldsymbol{X}; \boldsymbol{W}_Q, \boldsymbol{W}_K) = \mathrm{softmax}\left(\boldsymbol{X}\boldsymbol{W}_Q\boldsymbol{W}_k^\top\boldsymbol{X}^\top\right), \tag{3}$$

where $X \in \mathbb{R}^{n \times d}$ is the input of the self-attention layer, and it is clear that the QK transformation satisfies the structure defined in eq. (1), with $\boldsymbol{W}_2 = \boldsymbol{W}_Q$, $\boldsymbol{W}_i = \boldsymbol{W}_k^\top$ and $\sigma$ being the identity mapping. All those components therefore possess permutation symmetry, and fall within the scope of our theory. A key consequence of the permutation symmetry is the *permutation equivariance* of the learning algorithms, such as (stochastic) gradient descent and Adam, which we will take as the starting point of our theory.

**Critical Learning Rates.** Recent empirical studies suggest that neural networks exhibit qualitatively different learning dynamics under small versus large learning rates. Large learning rates often lead to simpler models (Galanti & Poggio, 2022; Chen et al., 2023; Dohare et al., 2024), while such dramatic changes seem to be lacking with the use of small learning rates, where the learning dynamics are often well-approximated by the NTK or the mean-field theories (Jacot et al., 2018; Yang & Hu, 2020; Mei et al., 2019). In this work, we establish a topological characterization of these transitions.

**Topology.** Topology is the mathematical study of abstract shapes and connectivity, focusing on properties that remain unchanged under continuous deformations such as stretching or bending (Kuratowski, 2014). It provides a way to talk about local and global structures without relying on exact distances; a bijective continuous map with a continuous inverse is called a homeomorphism and preserves the topology of general sets. Manifolds are sets with a local Euclidean structure, and their smooth structure is preserved under smooth invertible maps, called diffeomorphisms (Lang, 2012). In the context of deep learning, topological perspectives has been widely adopted to understand the properties of neural networks (Bucarelli et al., 2024; Barannikov et al., 2020; Horoi et al., 2022; Naitzat et al., 2020; Purvine et al., 2023; Nurisso et al., 2024; Birdal et al., 2021). However, prior studies have largely remained empirical and focused on specific networks, whereas our work connects the topology of neurons systematically to the training dynamics.

## 3 PRELIMINARIES

Now, we temporarily set aside considerations of specific neural networks and learning algorithms, and instead focus on more general and abstract objects. We will consider a (possibly infinite) collection of high-dimensional particles (corresponding to neurons) and their dynamics (corresponding to learning algorithms). In the following, we use the terms "particles" and "neurons" interchangeably.

**Notations.** Let $I$ be an arbitrary potentially uncountable set, which we often refer to as the index set. Throughout this paper, we focus on a collection of $D$-dimensional vectors indexed by $I$. We use $\left(\mathbb{R}^D\right)^I$ to represent the set of all such collections. We use calligraphic uppercase letters to denote collections indexed by $I$ (e.g. $\mathcal{X} \in \left(\mathbb{R}^D\right)^I$), bold lowercase letters to denote vectors (e.g. $\boldsymbol{x} \in \mathbb{R}^D$), and unbold lowercase letters to denote scalars or an entry of a vector or matrix (e.g. $x_k \in \mathbb{R}$ represents the $k$-th entry of $\boldsymbol{x}$). For $i \in I$, and a vector $\boldsymbol{v} \in \mathbb{R}^D$, we use $\boldsymbol{e}_i \cdot \boldsymbol{v} \in \left(\mathbb{R}^D\right)^I$ to denote a collection of $D$-dimensional vectors where only the $i$-th element is $\boldsymbol{v}$ and other vectors are $\boldsymbol{0}$.

Let $\mathsf{FSym}(I)$ be the Finitary Permutation Group on $I$, i.e. the group of all permutation operators on $I$ with a finite support (Neumann, 1976). For an operator $P \in \mathsf{FSym}(I)$ and $\mathcal{X} = \{\boldsymbol{x}_i\}_{i \in I}$, we use $P\mathcal{X}$ to represent $\left\{\boldsymbol{x}_{P(i)}\right\}_{i \in I}$.

For $\mathcal{X} = \{\boldsymbol{x}_i\}_{i \in I} \in \left(\mathbb{R}^D\right)^I$ and $\mathcal{Y} = \{\boldsymbol{y}_i\}_{i \in I} \in \left(\mathbb{R}^D\right)^I$, if $\mathcal{X}$ and $\mathcal{Y}$ only differs in finite many terms, then we define $\|\mathcal{X} - \mathcal{Y}\| = \sqrt{\sum_{i \in I} \|\boldsymbol{x}_i - \boldsymbol{y}_i\|^2}$, and $\|\mathcal{X} - \mathcal{Y}\| = +\infty$ if otherwise.

**Problem Setting.** Formally, we focus on a evolving collection of $D$-dimensional vectors

$$\mathcal{X}^{(t)} = \left\{\boldsymbol{x}_i^{(t)}\right\}_{i \in I} \in \left(\mathbb{R}^D\right)^I, \tag{4}$$

where $t \in \mathbb{N}$ is the time axis. $\mathcal{X}^{(t)}$ is updated by a generic update rule $U^{(t)} : \left(\mathbb{R}^D\right)^I \to \left(\mathbb{R}^D\right)^I$ with step size $\eta > 0$:

$$\boldsymbol{x}_i^{(t+1)} = \boldsymbol{x}_i^{(t)} + \eta U_i^{(t)}\left(\mathcal{X}^{(t)}\right), \tag{5}$$

where $U_i^{(t)}(\mathcal{X}) = \left(U^{(t)}(\mathcal{X})\right)_i$. Here, each element in $\mathcal{X}^{(t)}$ corresponds to the weights of a neuron of a neural network at training step $t$, and $U^{(t)}$ corresponds to the learning algorithm at time point $t$, which includes regularization terms, and can also depend on other parameters that are not considered (this is also why the update rule is time-dependent, as other parameters can change over time).

Abstractly, we consider update rules $U^{(t)}$ satisfying the following properties.

- **(P1) Equivariance Property**: We say $U^{(t)}$ has equivariance property if for any $t \in \mathbb{N}$, any $\mathcal{X} \in \left(\mathbb{R}^D\right)^I$ and $P \in \mathsf{FSym}(I)$, we have $PU^{(t)}(\mathcal{X}) = U^{(t)}(P\mathcal{X})$. In deep learning, this property

is a consequence of running gradient-based algorithms on permutation-symmetric loss functions, as we will show in Section 5. That many dynamics are naturally equivariant has been studied in physics Field (1980), but its role in deep learning is not yet clear.

- **(P2-$K$) $K$-Continuity Property**: For $K > 0$, if for any $t \in \mathbb{N}$ and any $\mathcal{X}, \mathcal{Y} \in \left(\mathbb{R}^D\right)^I$,

$$\left\| U^{(t)}\left(\mathcal{X}\right) - U^{(t)}\left(\mathcal{Y}\right) \right\| \le K \left\| \mathcal{X} - \mathcal{Y} \right\|, \tag{6}$$

  then we say $U^{(t)}$ has $K$-continuity property. Notice that eq. (6) makes sense only when $\mathcal{X}$ and $\mathcal{Y}$ only differ in finite entries. When $I$ is finite and $U^{(t)}$ is gradient descent, the quantity $K$ is the largest eigenvalue of the Hessian of the loss function, and the $K$-continuity property becomes an upper bound of the Lipschitz continuity of the gradient, which is commonly seen in optimization theory (See details in Section 5.1). We intentionally choose this form to establish this correspondence; however, it is possible to prove our theory with a weaker version of the continuity property. See Appendix D for details.

Now, it is important to define the word "neuron." The equivariance property is actually the most general way to define a "neuron" (e.g., see Ziyin (2024)), whatever subset of parameters that are permutationally-equivalent to the learning rule can be called a "neuron." In case of fully connected networks trained with GD, this definition of a "neuron" is equivalent to the standard definition (incoming plus outgoing weights of an activation unit).

If, beyond topology, we also want to talk about the differentiable manifold structure of the neurons, we further consider a smoothness property of $U$.

- **(P3) Smoothness Property**: For any $i \in I$ and $t \in \mathbb{N}$, define

$$\forall \boldsymbol{y}, \boldsymbol{z} \in \mathbb{R}^D, g_i^{(t)}(\boldsymbol{y}, \boldsymbol{z}) = U_i^{(t)}\left(\mathcal{X}^{(t)} + \left(\boldsymbol{e}_i - \boldsymbol{e}_j\right) \cdot \boldsymbol{z}\right), \text{ such that } \boldsymbol{y} = \boldsymbol{x}_j^{(t)}, \tag{7}$$

  where $j$ is arbitrarily chosen when multiple $j$-s satisfies the condition, and $\boldsymbol{e}_j$ is set to $\boldsymbol{0}$ if no $j$ satisfies the condition. If $g_i^{(t)}$ is $C^1$ on $\left(\mathbb{R}^D\right)^2$ for any $i \in I$, we say $U^{(t)}$ has the ($C^1$-)smoothness property. Intuitively, the smoothness property requires that the response of each output entry of $U$ with respect to a small perturbation of one entry of its input must be $C^1$.

## 4 TOPOLOGY OF LEARNING

In this section, we present our main theoretical results: the characterization of the change of topology and measure structures of $\mathcal{X}$ under the update rule. Before diving into the main theorems, we first establish two critical lemmas. These lemmas show that, a combination of the equivariance and continuity of the update rule implies that there is an emergent notion of distance between different neurons. Intuitively, permutation equivariance implies that two infinitesimally close neurons need to have identical updates, which implies that the motion that changes their difference must be vanishing. Thus, equivariance ensures that neurons that start close to each other remain close because dynamics that would increase or decrease their distance are suppressed.

**Lemma 1** (Well-definedness). *The following statement holds when $U^{(t)}$ satisfies P1. For any $i, j \in I$ such that $i \ne j$, if at time $t$ we have $\boldsymbol{x}_i^{(t)} = \boldsymbol{x}_j^{(t)}$, then, $\boldsymbol{x}_i^{(t+1)} = \boldsymbol{x}_j^{(t+1)}$.*

Next, we strengthen the intuition behind Lemma 1 by incorporating the continuity property.

**Lemma 2** (No Merging or Splitting). *If $U^{(t)}$ satisfies P1 and P2-K, then for any $i, j \in I$ such that $i \ne j$,*

$$(1 - \eta K) \left\| \boldsymbol{x}_i^{(t)} - \boldsymbol{x}_j^{(t)} \right\| \le \left\| \boldsymbol{x}_i^{(t+1)} - \boldsymbol{x}_j^{(t+1)} \right\| \le (1 + \eta K) \left\| \boldsymbol{x}_i^{(t)} - \boldsymbol{x}_j^{(t)} \right\|. \tag{8}$$

This lemma implies the bi-Lipschitzness of the update rule between the manifolds formed by neurons at consecutive time steps $t$ and $t + 1$. The fact that common learning rules induce bi-Lipschitz mappings is nontrivial, as such maps are known to preserve topological invariants (as we will show in the subsequent section) and control geometric distortions (Heinonen, 2001).

Moreover, Lemma 2 also identifies a critical learning rate $\eta^* = 1/K$, beyond which the lower bound becomes vacuous, which we referred to as *topological critical point* hereinafter. As we will see in the next section, this marks a phase transition from bijective, homeomorphic dynamics to merely surjective continuous dynamics.

## 4.1 TOPOLOGICAL INVARIANCE

A crucial perspective is implied by the lemmas above: the entirety of the neurons can be seen as a set (or, manifold) $S \subset \mathbb{R}^D$, and the evolution of neurons can be viewed as the evolution of $S$. Of course, a crucial question is whether such a perspective is meaningful, which is the key question we answer in this section.

Formally, let $S^{(t)} = \left\{ \boldsymbol{x}_i^{(t)} \,\middle|\, i \in I \right\} \subseteq \mathbb{R}^D$ denote the set formed by all of the neurons in $\mathcal{X}^{(t)}$, equipped with the relative topology inherited from $\mathbb{R}^D$. Define function $\widehat{U}^{(t)} : S^{(t)} \to S^{(t+1)}$ by

$$\forall x \in I, \widehat{U}^{(t)} \left( \boldsymbol{x}_i^{(t)} \right) = \boldsymbol{x}_i^{(t+1)}. \tag{9}$$

Intuitively, $\widehat{U}^{(t)}$ describes the effect of $U^{(t)}$ on each point of $S^{(t)}$. Lemmas 1 and 2 together ensure that $\widehat{U}^{(t)}$ is well-defined and, under small learning rates, a bijection.

**Lemma 3.** *If $U^{(t)}$ satisfies P1 and P2-K, then $\widehat{U}^{(t)}$ is well-defined, and is a surjection. If additionally $\eta K < 1$, then $\widehat{U}^{(t)}$ is a bijection.*

One can show that, $\widehat{U}^{(t)}$ is not only a bijective, but also a homeomorphism between $S^{(t)}$ and $S^{(t+1)}$. This leads to our main theorem.

**Theorem 1** (Main). *If $U^{(t)}$ satisfies P1 and P2-K, then*

  i. *$\widehat{U}^{(t)}$ is a continuous surjection from $S^{(t)}$ to $S^{(t+1)}$;*
  ii. *if $S^{(t)}$ is compact, then $S^{(t+1)}$ is also compact, and $\widehat{U}^{(t)}$ is a quotient map;*
  iii. *if $\eta K < 1$, then $\widehat{U}^{(t)}$ is a homeomorphism;*
  iv. *if $\eta K < 1$, and $U^{(t)}$ also satisfies P3, and $S^{(t)}$ is an open subset of $\mathbb{R}^D$, then $S^{(t+1)}$ is also open, and $\widehat{U}^{(t)}$ is a $C^1$-diffeomorphism.*

See Appendix A.4 for the proof of Theorem 1. This result shows that when the learning rate is below the critical threshold $\eta^* = 1/K$, the neuronal set $S^{(t)}$ evolves through homeomorphisms (or diffeomorphisms if smoothness holds). Consequently, the topology of $S^{(t)}$ remains invariant across training: if the neurons initially form a space homeomorphic to a circle, torus, or any other manifold, they will preserve that topological type for all time. If the neurons are initially separated points that are far away from each other, this statement has a simple interpretation: neurons cannot merge unless they were identical at initialization, whereas if two neurons are merged, they cannot be separated. This implies that the learning process can only locally deform the neuron topology, either by translating, expanding, or contracting local neuron densities.

That a sufficiently smooth learning rule induces a diffeomorphism of the neuron manifold both lends support to the widespread use of mean-field theories (including the NTK theory) for understanding neural networks training at a small learning rate (Mei et al., 2019; Jacot et al., 2018), and explains their breakdown at a large learning rate. The diffeomorphic evolution ensures that the neuron distribution $P_t(w)$ obeys standard change-of-variable formulas, leading to Vlasov-type equations in the infinite-width limit (Spohn, 2012). Since our theory is independent of the specific architecture of the neural network, it could lead to the most general type of mean-field theory for deep learning, which we leave as a future direction.

At large learning rates, by contrast, homeomorphic evolution breaks down. Merging and more general topological changes become possible so that the learning process can no longer be described as local interactions and the mean-field theories no longer apply. This transition, from topology-preserving to topology-changing dynamics, constitutes the *topological critical point* predicted by our theory and is verified in our experiments (Section 6). At the same time, the large-learning-rate phase cannot change topology without bound because the upper bound in Lemma 2 always holds, and so neuron splitting remains impossible. This is also topologically characterized by the fact that the induced mapping $\tilde{U}$ is still a quotient map, meaning that it inherits a *coarser* topology from the previous neuron distribution. The implication of reaching a coarser topology is that the training reduces the expressivity/capacity of these neural networks and therefore simplifies them. This can be understood directly from the perspective of permutation symmetries, where merging (or gluing) two neurons is the same as transitioning to the symmetric state of the permutation symmetry, which directly reduces the effective number of parameters of the model by the number of weights in a neuron (e.g., see Proposition 3 of Ziyin et al. (2025)).

**Role of $K$.** So far, we have formally treated the smoothness parameter $K$ as a global quantity, which leads to the elegant and easy-to-state results above. However, it is much better to conceptually treat $K$ as a local quantity in a small neighborhood around the current parameter $\theta$: $K \approx K(\theta)$. When the learning rule is SGD, this $K(\theta)$ can be approximated by the largest eigenvalue of the local Hessian $\lambda_{\max}(H(\theta))$. In this more dynamical perspective, $K$ can be seen as a dynamically evolving quantity. When viewed with the phenomenon of the edge of stability (Cohen et al., 2021; Wu et al., 2018), this picture suggests a two-phase perspective of the learning process of common neural networks, where the first phase of training focuses on optimizing the loss and learning the task, while the second phase of learning is a simplification process, where the model tends to simpler and coarser topologies, a process that could be related to phenomena such as grokking (Power et al., 2022).

### 4.2 ON A QUANTITATIVE DESCRIPTION

Theorem 1 presents a result regarding the mathematical topology of the neurons, which primarily addressing the case in which the number of neurons is considered infinite. In this section, we provide a more practical result that upper-bounds the scale change of the neurons. To formalize this, we first define the $r$-expansion of a set.

**Definition 1.** *For a set $P \subseteq \mathbb{R}^d$ and a scalar $r > 0$, the $r$-**expansion** of $P$ is defined as*

$$P^r = \left\{ y \in \mathbb{R}^d \middle| \|x - y\| < r \right\}. \tag{10}$$

Note that the $r$-expansion defined here coincides with the standard notion of expansion widely used in the study of metric spaces. For any set $P \subseteq \mathbb{R}^d$, its $r$-expansion is naturally an open set and inherits the relative topology from $\mathbb{R}^d$.

**Theorem 2.** *If $U^{(t)}$ satisfies P1 and P2-K and $\eta K < 1$, then for any $r < \inf_{x,y \in S^{(t)}} \|x - y\|$, we have $\left(S^{(t)}\right)^r$ is homeomorphic to $\left(S^{(t+1)}\right)^{(1-\eta K)r}$.*

See Appendix A.9 for the proof of Theorem 2. Note that when $r = 0$, Theorem 2 reduces to Theorem 1.iii. When $r > 0$, however, it provides an quantitative description of how the scale of the set changes.

### 4.3 MEASURE INVARIANCE

Beyond the invariance of topology, one can also ask "how many" neurons are stacked at a single point of $S^{(t)}$ and how their density evolves over time. Formally speaking, this corresponds to studying the probability distribution on $S^{(t)}$ obtained by pushing forward a universal probability distribution defined on the index set $I$. In this subsection, we show that this distribution is also preserved under the update rule.

Formally, in this subsection we assume a concrete structure on the index set $I$. Assume there is a $\sigma$-algebra $\mathcal{F}$ on $I$ and a probability measure $m : \mathcal{F} \to [0, 1]$. At any time $t \in \mathbb{N}$, define the mapping $r^{(t)} : i \mapsto x_i^{(t)}$, and let it be a measurable function from $I$ to $S^{(t)}$, where $S^{(t)}$ carries the corresponding Borel $\sigma$-algebra.[2] For each time $t$, we define a measure $\mu^{(t)}$ on $S^{(t)}$ as the push-forward of $m$ under $r^{(t)}$, i.e.

$$\forall \text{open set } A \text{ on } S^{(t)}, \mu^{(t)}(A) = m\left(r^{(t)^{-1}}(A)\right). \tag{11}$$

Clearly, $\mu^{(t)}$ is also a probability measure.

**Theorem 3.** *Suppose $U^{(t)}$ satisfies P1 and P2-K, and suppose $\eta K < 1$, then $\widehat{U}^{(t)}$ is a probability isomorphism between $\left(S^{(t)}, \mu^{(t)}\right)$ and $\left(S^{(t+1)}, \mu^{(t+1)}\right)$, i.e. $\widehat{U}^{(t)}$ and $\widehat{U}^{(t)^{-1}}$ are both measure-preserving bijections.*

Theorem 3 shows that the update rule preserves not only the topology of $S^{(t)}$, but also the density of neurons across it. Theorem 1 and Theorem 3 might remind readers of the topological dynamical

---

[2]These assumptions are automatically satisfied with a finite $I$.

systems and measure-preserving dynamical systems (Gottschalk & Hedlund, 1955). However, what we study is more general because in this context the update rule $\widehat{U}^{(t)}$ as well as the space $S^{(t)}$ itself are both time-dependent, which violates the definition of the topological/measure-preserving dynamical systems. Therefore, the classical recurrence theorems in those fields cannot be directly applied.

## 5  EXAMPLES: GRADIENT DESCENT AND ADAM

Starting from this section, we connect our abstract theoretical discussions to actual optimization algorithms, by proving that the properties of the update rule $U^{(t)}$ we have used are satisfied by a wide range of optimization algorithms. Here we analyze two of the most popular ones, namely Gradient Descent (GD) and Adam, as illustrative cases.

### 5.1  GRADIENT DESCENT

Let us assume that there is a loss function $L : \left(\mathbb{R}^D\right)^I \to \mathbb{R}$ that maps the neurons to a scalar loss value, and the update rule $U^{(t)}$ is the gradient descent update[3]:

$$U^{(t)}(\mathcal{X}) = -\nabla L(\mathcal{X}). \tag{12}$$

Now we prove that, the equivariance property of $U$ comes from the permutation symmetry of $L$ and the continuity property comes from the smoothness of $L$.

**Proposition 1.** *If $L$ has $\mathsf{FSym}(I)$-symmetry, i.e.*

$$\forall \mathcal{X} \in \left(\mathbb{R}^D\right)^I, \forall P \in \mathsf{FSym}(I), L(\mathcal{X}) = L(P\mathcal{X}), \tag{13}$$

*then $U^{(t)}$ defined in eq. (12) satisfies P1.*

**Proposition 2.** *If there exists a constant $K > 0$, such that for any $\mathcal{X}, \mathcal{Y} \in \left(\mathbb{R}^D\right)^I$ and any $i \in I$, we have*

$$\left\| \nabla L\left(\mathcal{X}\right) - \nabla L\left(\mathcal{Y}\right) \right\| \le K \left\| \mathcal{X} - \mathcal{Y} \right\|, \tag{14}$$

*then $U^{(t)}$ defined in eq. (12) satisfies P2-$K$.*

**Remark.** As noted in Section 3, the smoothness condition in eq. (14) is precisely the standard smoothness assumption widely used in optimization theory (Bottou et al., 2018). Under this assumption, the topological critical point $\eta^* = \frac{1}{K}$ in our theory coincides with the optimal step size suggested by a second-order Taylor expansion of the loss around $\boldsymbol{x}$. Specifically, eq. (14) implies

$$L(\boldsymbol{x} - \eta \nabla L(\boldsymbol{x})) \tag{15}$$

$$\le L(\boldsymbol{x}) - \eta \left\| \nabla L(\boldsymbol{x}) \right\|^2 + \frac{K\eta^2}{2} \left\| \nabla L(\boldsymbol{x}) \right\|^2 \tag{16}$$

$$= L(\boldsymbol{x}) + \left(\frac{K}{2}\eta^2 - \eta\right) \left\| \nabla L(\boldsymbol{x}) \right\|^2. \tag{17}$$

It follows that the optimal decrease in loss occurs at $\eta^* = \frac{1}{K}$, which matches the topological critical point identified in our framework and differs only by a constant factor from the classical upper bound on stable

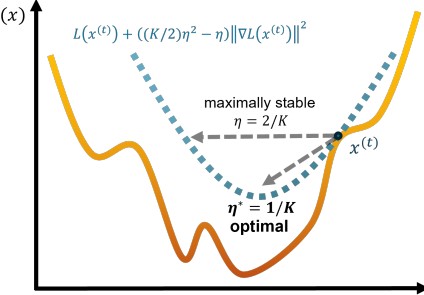

Figure 2: **An optimization perspective of the topological critical point**. The topological critical point $\eta^* = 1/K$ corresponds to the step size that reduces the loss optimally, while the critical step size found by Cohen et al. (2021) corresponds to the largest one ensuring loss decay.

step sizes. See Figure 2 for an illustration. This correspondence suggests there might be a hidden connection between neuron topology and optimization, under the context of gradient descent and the presence of permutation symmetry: *The loss can be stably optimized only when the topology of the neurons is preserved.*

---

[3]For this case, the right-hand side of eq. (12) is independent of time $t$, and therefore $U^{(t)}$ is the same at each time. However, we would love to keep this redundancy of notation, because here we have actually made a subtle (but harmless) simplification, that we implicitly assume all neurons that are to be updated have permutation symmetry, while in practice there can be a part of learnable parameters that are not permutation-invariant, absorbing which into $L$, although does not affect our discussion here, will make the loss function $L$ time-dependent and so does the update rule.

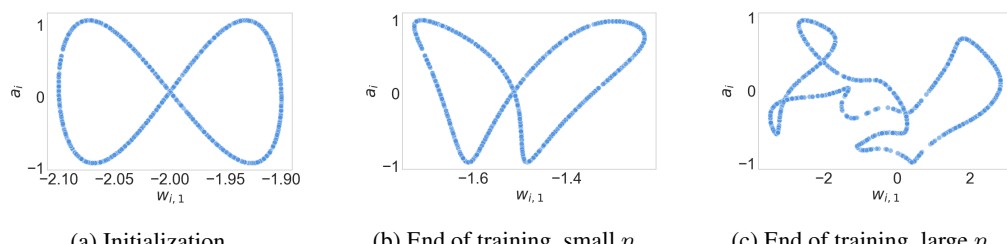

(a) Initialization      (b) End of training, small $\eta$      (c) End of training, large $\eta$

Figure 3: **Topology of a 2D neural network with GD.** The neurons are initialized on a genus-2 surface and optimized with GD. We visualize the topology of 2D and 3D networks before and after training under different step sizes $\eta$. For small step sizes, the training may deform the geometric arrangement of the neurons but the topology remains unchanged. In contrast, for large step sizes, the topological structure can change substantially. These results consistently verify our theoretical predictions that while the geometry of the neurons can be affected by training, the underlying topology is stable under small learning rates but fragile under large ones.

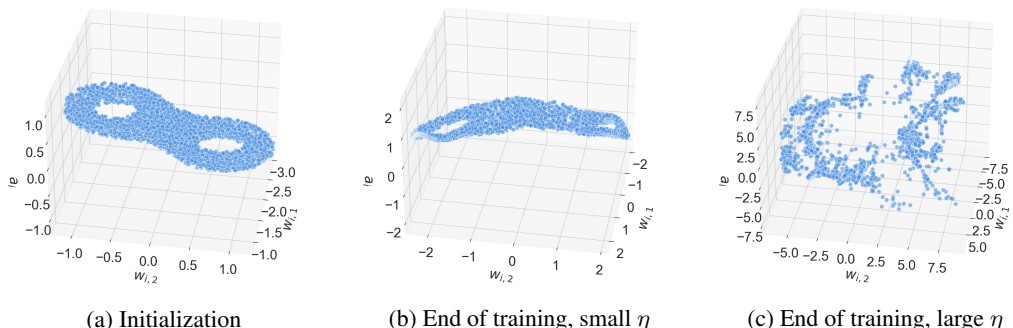

(a) Initialization      (b) End of training, small $\eta$      (c) End of training, large $\eta$

Figure 4: **Topology of a 3D neural network with GD.** The neurons are initialized on a genus-2 surface and optimized with GD.

## 5.2 ADAM

Besides GD, other more complicated and modern optimizers are usually stateful – they need to keep track of some values in the process of training, which at first sight seems incompatible with our definition of $U^{(t)}$, since our $U^{(t)}$ is stateless by definition. This seemingly difficulty can be resolved by a small trick: we can view the state of an optimizer as a part of the neurons, thereby rewriting the update rule in a stateless form. As an illustration, in this section we prove that Adam (Kingma & Ba, 2015), another widely-used optimizer in deep learning, also fits in our framework.

Specifically, suppose $\boldsymbol{\theta}_i^{(t)}$ is the $i$-th neuron in the neural network at time $t$, the collection of particles is defined as $\mathcal{X}^{(t)} = \left\{ \left( \boldsymbol{\theta}_i^{(t)}, \boldsymbol{m}_i^{(t)}, \boldsymbol{v}_i^{(t)} \right) \right\}_{i \in I}$, where $\boldsymbol{m}_i^{(t)}$ and $\boldsymbol{v}_i^{(t)}$ are the first order and second order moment estimators in Adam. The update rule is then defined as[4]

$$U^{(t)} : \left\{ \begin{pmatrix} \boldsymbol{\theta}_i \\ \boldsymbol{m}_i \\ \boldsymbol{v}_i \end{pmatrix} \right\}_{i \in I} \mapsto \left\{ \begin{pmatrix} -\frac{\boldsymbol{m}_i/(1-\beta_1^t)}{\epsilon + \sqrt{\boldsymbol{v}_i/(1-\beta_2^t)}} \\ \frac{1-\beta_1}{\eta} \left[ \nabla_i L(\Theta) - \boldsymbol{m}_i \right] \\ \frac{1-\beta_2}{\eta} \left[ \nabla_i L(\Theta)^2 - \boldsymbol{v}_i \right] \end{pmatrix} \right\}_{i \in I}, \tag{18}$$

where $\beta_1$ and $\beta_2$ are the decay rates for the first- and second-order moment estimators, respectively; $\Theta = \{\boldsymbol{\theta}_i\}_{i \in I}$ is the collection of the neurons; and all scalar operations (square, division, square root) are taken element-wise. It is straightforward to check that the neuron update in eq. (18) is equivalent to the standard Adam rule. One can now prove the following theorem.

**Proposition 3.** *If $L$ has* $\mathsf{FSym}(I)$*-symmetry (eq. (13)), then $U^{(t)}$ defined in eq. (18) satisfies P1.*

## 6 EXPERIMENTS

To illustrate our theoretical results, we conduct experiments using gradient-based methods on real neural networks and track changes in the topological structure of the neuron-induced point cloud.

---

[4]Here and in the appendix, we optionally write the tuple in the tall form for better presentation.

Our experimental results include both direct visualizations of the topological structure in low-dimensional networks and quantitative measurements that capture topological properties in networks trained on standard tasks. The experiments are conducted under a variety of optimizers and settings. Additional results and detailed experimental settings are in Appendices B and C.

**Low-dimensional Distributions.** We first train neural networks with low-dimensional neurons (2- or 3-dimensional) in order to directly visualize their topology. Specifically, consider a two-layer neural network $F : \mathbb{R}^d \to \mathbb{R}$ with hidden layer size $h$, defined as

$$F\left(\boldsymbol{z}; \{(\boldsymbol{w}_i, a_i)\}_{i=1}^h\right) = \sum_{i=1}^h a_i \sigma\left(\langle \boldsymbol{w}_i, \boldsymbol{z}\rangle\right), \tag{19}$$

where $\boldsymbol{w}_i \in \mathbb{R}^d$ and $a_i \in \mathbb{R}$ are learnable parameters, and $\sigma$ denotes the sigmoid function. The network is trained on data generated by a random teacher network (See Appendix C for details). In this setting, the loss function has the permutation symmetry as described in eq. (13), with $I = [h]$ and $\mathcal{X} = \{(\boldsymbol{w}_i, a_i)\}_{i \in I} \in (\mathbb{R}^{d+1})^I$.

For visualization purposes, we focus on $d = 1$ and $d = 2$, so that each element in $\mathcal{X}$ lies in $\mathbb{R}^2$ or $\mathbb{R}^3$ (referred to below as 2D and 3D networks, respectively), which enables a straightforward visualization of their topology. Moreover, we initialize the elements in $\mathcal{X}$ with specific topological structures to highlight potential topological (in)variance. See Figures 3, 4 and 6 for the results with GD, and Appendix B for extra results with other optimizers.

**Topological Invariants.** Now, we directly measure topological invariants of high-dimensional models trained on real tasks. Specifically, we measure the first three Betti numbers $b_0, b_1, b_2$ of the point cloud formed by the neurons. Betti numbers are fundamental topological invariants that count the number of connected components, loops, and higher-dimensional voids in a topological space; they are widely used in topological data analysis as compact descriptors of shape and structure (Edelsbrunner & Harer, 2010; Naitzat et al., 2020).

We train a two-layer MLP on the MNIST dataset (LeCun, 1998) for a classification task using standard cross-entropy loss, and track the evolution of Betti numbers. The network is initialized with neurons uniformly sampled from the surface of a 3D unit sphere, which has Betti numbers $(b_0, b_1, b_2) = (1, 0, 1)$. Figure 5 shows the results with both GD and Adam. These results are consistent with our theoretical predictions: with small learning rates, the model learns without changing the topological structure of the neurons, while with large learning rates, the topology can change. Importantly, in all cases the model can achieve a significant test accuracy, ruling out the possibility that with small step sizes the model simply stays near initialization without meaningful updates.

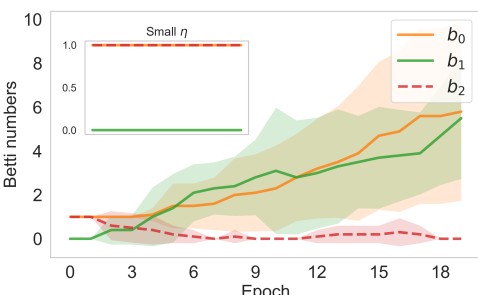

Figure 5: **Evolution of Betti numbers during training with GD.** The main panel shows results for the large learning rate, while the inset shows results for the small one. Each curve is obtained by averaging over 10 runs; the shaded regions indicate the standard deviations.

## 7 DISCUSSION

We have investigated the interplay between permutation symmetries, learning rates, and neuron topology in the training dynamics of neural networks, leading to a universal conceptual message: permutation symmetry of architecture modules or learning algorithms imposes strong topological constraints on how learning could happen. These interactions yield a range of insights that shed light on understanding important empirical phenomena and inspire future algorithm design. A limitation of our work is that it is entirely theoretical and does not test the predictions on large-scale experiments. Due to the scope limit, we have also only discussed a small subset of all possible implications of a topological theory of deep learning.

**Topology.** Our results establish a simple and clear topological characterization of learning, and clarify a crucial distinction between training with different learning rates: small learning rates preserve the topological structure of the neuron manifold, whereas large learning rates may enable

topological changes. This might provide new insights into learning rate scheduling strategies, such as learning rate decay: starting with a relatively large learning rate may facilitate exploration across different topological configurations, while subsequent decay to smaller values can stabilize the training dynamics within a fixed topology. Our theory also offers a structural viewpoint that complements existing explanations, such as the "catapult mechanism" (Lewkowycz et al., 2020). While further work is needed to establish the precise conditions under which such topological transitions occur in practical settings, this perspective highlights a potentially useful link between learning-rate phases and topological dynamics.

**Phase Transition.** From a physics perspective, the change in the topology directly corresponds to phase transitions. For example, a material with different Chern numbers is in different phases. In our setting, these topological phase transitions also directly correspond to changes in the symmetry of the parameters and are thus also phase transitions of the Landau type. Specifically, changing from a genus-1 topology to a genus-2 topology implies that two neurons have "merged" into one neuron, and this corresponds to a symmetry-restoration process where the network changes from the symmetry-broken state to the symmetric state (Ziyin, 2024).

**Deep Learning Theory.** Our result also highlights the limitations of conventional theories of learning dynamics. The EOS phenomenon states that GD almost always leads to a solution whose sharpness is $2/\eta$, and in practice this can happen quite early on in the training. Our result thus suggests a huge difference between dominant theories of learning dynamics such as NTK and mean-field theories, and the actual learning dynamics that we observe in practice. The topological breakdown implies that the theories built for a smaller learning rate cannot approximate what happens above that critical point, and it remains an open problem of how to describe the learning processes in the topological breakdown regime.

**Interdisciplinary link to Neuroscience.** An important line of thought in neuroscience is to understand our brain, the biological collection of neurons, as a manifold, whose topological and geometrical properties encode information (Perich et al., 2025). It is no coincidence that artificial neural networks are used and identified as mathematical models of the brain (for example, the cerebellum is often modeled as a fully connected feedforward network (Xie et al., 2023)). Therefore, our work may be further extended to help us understand the biological brain and advance neuroscience.

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

# A  PROOFS OF THEORETICAL RESULTS

In this section, we prove all the theoretical results in the main paper. Before starting, we first define some additional notation. For a collection, we use subscripts to denote its elements. For example, if $\mathcal{X} = \{\boldsymbol{x}_i\}_{i \in I} \in \left(\mathbb{R}^D\right)^I$ is a collection of $D$-dimensional vectors, then $\mathcal{X}_i$ represents $\boldsymbol{x}_i$ by default.

For an operator $P$ on $I$, and a subset $J \subseteq I$, we use $P_J$ to denote the operator obtained by constraining $P$ on $J$.

For a statement $\psi$, we use $\mathbb{1}_{\{\psi\}}$ to represent its indicator, i.e. $\mathbb{1}_{\{\psi\}} = \begin{cases} 1 & \psi \text{ is true} \\ 0 & \text{otherwise} \end{cases}$.

## A.1  PROOF OF LEMMA 1

Let $\mathcal{X} = \mathcal{X}^{(t)}$ for convenience. Define $P : I \to I$ as switching $i$ and $j$:

$$\forall k \in I, P(k) = \begin{cases} j & k = i \\ i & k = j \\ k & \text{otherwise} \end{cases}. \tag{20}$$

Obviously $P \in \mathsf{FSym}(I)$. Since $\boldsymbol{x}_i^{(t)} = \boldsymbol{x}_j^{(t)}$, we have $\mathcal{X} = P\mathcal{X}$.

Applying $\mathcal{X} = P\mathcal{X}$ and the equivariance property, we can obtain that

$$\boldsymbol{x}_i^{(t+1)} = \boldsymbol{x}_i^{(t)} + \eta U_i(\mathcal{X}) = \boldsymbol{x}_j^{(t)} + \eta U_i(P\mathcal{X}) = \boldsymbol{x}_j^{(t)} + \eta (PU(\mathcal{X}))_i = \boldsymbol{x}_j^{(t)} + \eta U_j(\mathcal{X}) = \boldsymbol{x}_j^{(t+1)}, \tag{21}$$

which proves the proposition.

## A.2  PROOF OF LEMMA 2

We first prove a lemma showing that two neurons that are close must remain close.

**Lemma 4** (No Splitting). *The following statement holds when $U$ has the equivariance property and $K$-continuity property. For any $t \in \mathbb{N}$ and $i, j \in I$ such that $i \neq j$, we have*

$$\left\| U_i(\mathcal{X}^{(t)}) - U_j(\mathcal{X}^{(t)}) \right\| \leq K \left\| \boldsymbol{x}_i^{(t)} - \boldsymbol{x}_j^{(t)} \right\|. \tag{22}$$

*Proof.* Let $\mathcal{X} = \mathcal{X}^{(t)}$ for convenience. Define $P \in \mathsf{FSym}(I)$ as switching $i$ and $j$ (as in eq. (20)). Then using the equivariance property, we have

$$U_i(\mathcal{X}) - U_j(\mathcal{X}) = U_i(\mathcal{X}) - (PU(\mathcal{X}))_i \tag{23}$$
$$= U_i(\mathcal{X}) - U_i(P\mathcal{X}). \tag{24}$$

Notice that $\mathcal{X}$ and $P\mathcal{X}$ only differ in entries $i$ and $j$. Using the $K$-continuity property, we have

$$\sqrt{2} \left\| U_i(\mathcal{X}) - U_j(\mathcal{X}) \right\| = \sqrt{\left\| U_i(\mathcal{X}) - U_j(\mathcal{X}) \right\|^2 + \left\| U_i(\mathcal{X}) - U_j(\mathcal{X}) \right\|^2} \tag{25}$$
$$= \sqrt{\left\| U_i(\mathcal{X}) - U_i(P\mathcal{X}) \right\|^2 + \left\| U_j(\mathcal{X}) - U_j(P\mathcal{X}) \right\|^2} \tag{26}$$
$$\leq \left\| U(\mathcal{X}) - U(P\mathcal{X}) \right\| \tag{27}$$
$$\leq K \left\| \mathcal{X} - P\mathcal{X} \right\| \tag{28}$$
$$= \sqrt{2} K \left\| \boldsymbol{x}_i^{(t)} - \boldsymbol{x}_j^{(t)} \right\|. \tag{29}$$

The proposition is thus proved by shifting the terms. $\square$

Next, we prove Lemma 2 using Lemma 4.

**Proof of Lemma 2**  Notice that

$$\left\| \boldsymbol{x}_i^{(t+1)} - \boldsymbol{x}_j^{(t+1)} \right\| = \left\| \boldsymbol{x}_i^{(t)} - \boldsymbol{x}_j^{(t)} + \eta \left( U_i(\mathcal{X}^{(t)}) - U_j(\mathcal{X}^{(t)}) \right) \right\| \tag{30}$$

$$\in \left\| \boldsymbol{x}_i^{(t)} - \boldsymbol{x}_j^{(t)} \right\| + [-\eta, +\eta] \times \left\| U_i(\mathcal{X}^{(t)}) - U_j(\mathcal{X}^{(t)}) \right\|. \tag{31}$$

The proposition is thus directly proved by applying Lemma 4. $\qquad\square$

### A.3  PROOF OF LEMMA 3

For simplicity in the proof we fix a time $t$ and denote $\widehat{U}^{(t)}$ by $f$. From Lemma 1, we have if $\boldsymbol{x}_i^{(t)} = \boldsymbol{x}_j^{(t)}$ then $\boldsymbol{x}_i^{(t+1)} = \boldsymbol{x}_j^{(t+1)}$, therefore $f$ is well-defined. Moreover, from the definition of $S^{(t)}$ and $S^{(t+1)}$, it is obvious that $f$ is a surjection.

Now suppose $\eta K < 1$. If $\boldsymbol{x}_i^{(t)} \neq \boldsymbol{x}_j^{(t)}$, the left-hand-side of Lemma 2 and the condition that $\eta K < 1$ together shows that $\boldsymbol{x}_i^{(t+1)} \neq \boldsymbol{x}_j^{(t+1)}$, following from which we have $f$ is an injection. Therefore, $f$ is a bijection.

### A.4  PROOF OF THEOREM 1

We fix a time point $t$ and denote $\widehat{U}^{(t)}$ by $f$. Lemma 3 has already proved that $f$ is a surjection. In this proof, we first prove the topological properties (i., ii. and iii.), and then the differentiable manifold property (iv.).

**Topological properties.**  For any pair of two different points $\boldsymbol{x}_i^{(t)}$ and $\boldsymbol{x}_j^{(t)}$, from the right-hand-side of Lemma 2, we have

$$\left\| f\left( \boldsymbol{x}_i^{(t)} \right) - f\left( \boldsymbol{x}_j^{(t)} \right) \right\| \le (1 + \eta K) \left\| \boldsymbol{x}_i^{(t)} - \boldsymbol{x}_j^{(t)} \right\|, \tag{32}$$

which shows that $f$ is $(1 + \eta K)$-Lipschitz continuous. Since all Lipschitz continuous functions are continuous, we have $f$ is also continuous.

If, additionally, $S^{(t)}$ is compact, then from the continuity of $f$ we immediately know $S^{(t+1)} = f\left( S^{(t)} \right)$ is compact. Moreover, since $S^{(t+1)}$ is a metric space, it is automatically Hausdorff, and it is known that a surjective mapping from a compact space to a Hausdorff space is a quotient map.

Now, suppose $\eta K < 1$ (without the compactness of $S^{(t)}$). Lemma 3 has proved that $f$ is a bijection. Consider the inversion of $f$. Let $g = f^{-1}$. It is obvious that for any $i \in I$, we have $g\left( \boldsymbol{x}_i^{(t+1)} \right) = \boldsymbol{x}_i^{(t)}$. Using the left-hand-side of Lemma 2, we have

$$\left\| g\left( \boldsymbol{x}_i^{(t+1)} \right) - g\left( \boldsymbol{x}_j^{(t+1)} \right) \right\| \le \frac{1}{1 - \eta K} \left\| \boldsymbol{x}_i^{(t+1)} - \boldsymbol{x}_j^{(t+1)} \right\| \tag{33}$$

for any $i, j \in I$, and therefore $g = f^{-1}$ is also continuous. This proves that $f$ is a homeomorphism.

**Differentiable manifold properties.**  Now, with the condition that $\eta K < 1$ and $U^{(t)}$ satisfies the smoothness property (P3), we prove that $f$ is a diffeomorphism from $S^{(t)}$ to $S^{(t+1)}$. The Invariance of Domain Theorem (See e.g. Theorem 2B.3 in Hatcher (2002)) guarantees that $S^{(t+1)}$ is also an open set in $\mathbb{R}^D$. Therefore, we only need to prove that $f$ and its inverse both have continuous derivatives.

Fix a point $\boldsymbol{x}_i^{(t)} \in S^{(t)}$. Since $S^{(t)}$ is open, there must be a scalar $r_i > 0$, such that for any $\boldsymbol{\Delta} \in \mathbb{R}^d$ with $\|\boldsymbol{\Delta}\| \le r_i$, we have $\boldsymbol{x}_i^{(t)} + \boldsymbol{\Delta} \in S^{(t)}$. Consider such a perturbation $\boldsymbol{\Delta}$, then there must be a $j \in I$ such that $\boldsymbol{x}_j^{(t)} = \boldsymbol{x}_i^{(t)} + \boldsymbol{\Delta}$. Let $P \in \mathsf{FSym}(I)$ be the permutation operator that exchanges $i$ and $j$ (as

defined in eq. (20)). We have

$$f(\boldsymbol{x}_i^{(t)} + \boldsymbol{\Delta}) = f\left(\boldsymbol{x}_j^{(t)}\right) \tag{34}$$

$$= U_j\left(\mathcal{X}^{(t)}\right) \tag{35}$$

$$= \left(PU\left(\mathcal{X}^{(t)}\right)\right)_i \tag{36}$$

$$= U_i\left(P\mathcal{X}^{(t)}\right) \qquad \text{(Equivariance Property)} \tag{37}$$

$$= U_i\left[\mathcal{X}^{(t)} + (\boldsymbol{e}_i - \boldsymbol{e}_j)\left(\boldsymbol{x}_j^{(t)} - \boldsymbol{x}_i^{(t)}\right)\right] \tag{38}$$

$$= g_i^{(t)}\left(\boldsymbol{\Delta}, \boldsymbol{x}_j^{(t)}\right) \tag{39}$$

$$= g_i^{(t)}\left(\boldsymbol{\Delta}, \boldsymbol{x}_i^{(t)} + \boldsymbol{\Delta}\right) \tag{40}$$

Since from P3 we know $g_i^{(t)}$ is $C^1$ with respect to its two parameters, from the chain rule we have $g_i^{(t)}\left(\boldsymbol{\Delta}, \boldsymbol{x}_i^{(t)} + \boldsymbol{\Delta}\right)$ is also $C^1$ with respect of $\boldsymbol{\Delta}$, and therefore $f$ is also $C^1$ at point $\boldsymbol{x}_i^{(i)}$. Since $i$ is arbitrarily chosen, $f$ is thus $C^1$ on entire $S^{(t)}$.

Next, we prove that $f^{-1}$ is also $C^1$. Again consider $\boldsymbol{x}_i^{(t)} \in S^{(t)}$. Since we already know $f$ is $C^1$, let its gradient at point $\boldsymbol{x}_i^{(t)}$ be $\boldsymbol{G}$ and we have for any unit vector $\boldsymbol{v}$, the directional derivative satisfies

$$\boldsymbol{G}\boldsymbol{v} = \lim_{\substack{\delta \to 0 \\ \delta \neq 0}} \frac{f\left(\boldsymbol{x}_i^{(t)} + \delta\boldsymbol{v}\right) - f\left(\boldsymbol{x}_i^{(t)}\right)}{\delta}. \tag{41}$$

Let $\alpha = 1 - \eta K, \beta = 1 + \eta K$. From Lemma 2, for any $\delta < r_i$ we have

$$\alpha \leq \frac{\left\|f\left(\boldsymbol{x}_i^{(t)} + \delta\boldsymbol{v}\right) - f\left(\boldsymbol{x}_i^{(t)}\right)\right\|}{|\delta|} \leq \beta. \tag{42}$$

Subtracting the bounds into eq. (41), we get

$$\|\boldsymbol{G}\boldsymbol{v}\| \in [\alpha, \beta], \tag{43}$$

which further implies that all singular-values of $\boldsymbol{G}$ are in $[\alpha, \beta]$, which means $\boldsymbol{G}$ is invertible. Since $\widehat{U}$ is $C^1$, inverse function theorem therefore shows $f^{-1}$ is also $C^1$.

### A.5 PROOF OF THEOREM 3

We fix a time point $t$ and denote $\widehat{U}^{(t)}$ by $f$. Lemma 3 has already proved that $f$ is a bijection. For any open set $A \subseteq S^{(t+1)}$, we have

$$\mu^{(t+1)}(A) = \mu^{(t+1)}\left\{\boldsymbol{x}_i^{(t+1)} \middle| i \in I, \boldsymbol{x}_i^{(t+1)} \in A\right\} \tag{44}$$

$$= m\left\{i \in I \middle| \boldsymbol{x}_i^{(t+1)} \in A\right\} \tag{45}$$

$$= m\left\{i \in \middle| f\left(\boldsymbol{x}_i^{(t)}\right) \in A\right\} \tag{46}$$

$$= m\left\{i \in \middle| \boldsymbol{x}_i^{(t)} \in f^{-1}(A)\right\} \tag{47}$$

$$= \mu^{(t)}\left(f^{-1}(A)\right). \tag{48}$$

This proves that $f$ is measure-preserving. Following the same process one can easily prove that $f^{-1}$ is also measure-preserving.

### A.6 PROOF OF PROPOSITION 1

In this proof we prove a slightly stronger version of the proposition originally stated in Proposition 1, without using the condition that $I$ is finite. The result for finite $I$ is thus a direct corollary.

We only need to prove that for any $\mathcal{X} \in \left(\mathbb{R}^D\right)^I$ and $P \in \mathsf{FSym}(I)$, we have

$$P\nabla L(\mathcal{X}) = \nabla L(P\mathcal{X}). \tag{49}$$

Now we fix $P \in \mathsf{FSym}(I)$ and consider any $\mathcal{X} \in \left(\mathbb{R}^D\right)^I$. Let

$$J = \{i \in I | Pi \neq i\}, \tag{50}$$

be the support set of $P$. Since $P$ is finitary, $J$ is a finite set. Therefore, we only need to prove the proposition of entries in $J$. define $L_J : \left(\mathbb{R}^D\right)^J \to \mathbb{R}$ such that

$$\forall \mathcal{Y} = \{\boldsymbol{y}_j\}_{j \in J}, L_J(\mathcal{Y}) = L\left(\left\{\mathbb{1}_{\{i \in J\}}\boldsymbol{y}_i + \mathbb{1}_{\{i \notin J\}}\boldsymbol{x}_i\right\}_{i \in I}\right). \tag{51}$$

The symmetry gives us

$$\forall \mathcal{Y} \in \left(\mathbb{R}^D\right)^J, L_J(\mathcal{Y}) = L_J(P|_J \mathcal{Y}), \tag{52}$$

where $P_J = P|_J$ is restriction of $P$ on $J$. Taking derivative of both sides gives

$$\nabla L_J(\mathcal{Y}) = P_J^\top \nabla L_J(P_J \mathcal{Y}), \tag{53}$$

shifting the terms and eq. (49) is proved.

## A.7   PROOF OF PROPOSITION 2

The proposition is directly proved by noticing that

$$\left\| U^{(t)}(\mathcal{X}) - U^{(t)}(\mathcal{Y}) \right\| = \left\| \nabla L(\mathcal{Y}) - \nabla L(\mathcal{X}) \right\| \leq K \|\mathcal{Y} - \mathcal{X}\|. \tag{54}$$

## A.8   PROOF OF PROPOSITION 3

We use eq. (49) proved before. Let $P \in \mathsf{FSym}(I)$. We have

$$U^{(t)}\left[\left\{\begin{pmatrix} \boldsymbol{\theta}_{P(i)} \\ \boldsymbol{m}_{P(i)} \\ \boldsymbol{v}_{P(i)} \end{pmatrix}\right\}_{i \in I}\right] = \left\{\begin{pmatrix} -\frac{\boldsymbol{m}_{P(i)}/(1-\beta_1^t)}{\epsilon+\sqrt{P\boldsymbol{v}_{P(i)}/(1-\beta_2^t)}} \\ \frac{1-\beta_1}{\eta}\left[\nabla_i L(P\Theta) - \boldsymbol{m}_{P(i)}\right] \\ \frac{1-\beta_2}{\eta}\left[\nabla_i L(P\Theta)^2 - \boldsymbol{v}_{P(i)}\right] \end{pmatrix}\right\}_{i \in I} \tag{55}$$

$$= \left\{\begin{pmatrix} -\frac{\boldsymbol{m}_{P(i)}/(1-\beta_1^t)}{\epsilon+\sqrt{P\boldsymbol{v}_{P(i)}/(1-\beta_2^t)}} \\ \frac{1-\beta_1}{\eta}\left[\nabla_{P(i)} L(\Theta) - \boldsymbol{m}_{P(i)}\right] \\ \frac{1-\beta_2}{\eta}\left[\nabla_{P(i)} L(\Theta)^2 - \boldsymbol{v}_{P(i)}\right] \end{pmatrix}\right\}_{i \in I} \tag{56}$$

$$= P\left\{\begin{pmatrix} -\frac{\boldsymbol{m}_i/(1-\beta_1^t)}{\epsilon+\sqrt{P\boldsymbol{v}_i/(1-\beta_2^t)}} \\ \frac{1-\beta_1}{\eta}\left[\nabla_i L(\Theta) - \boldsymbol{m}_i\right] \\ \frac{1-\beta_2}{\eta}\left[\nabla_i L(\Theta)^2 - \boldsymbol{v}_i\right] \end{pmatrix}\right\}_{i \in I} \tag{57}$$

$$= PU^{(t)}\left[\left\{\begin{pmatrix} \boldsymbol{\theta}_i \\ \boldsymbol{m}_i \\ \boldsymbol{v}_i \end{pmatrix}\right\}_{i \in I}\right]. \tag{58}$$

## A.9   PROOF OF THEOREM 2

Let $\delta = \inf_{x,y \in S^{(t)}}$. When $\delta = 0$, the result is a direct corollary of Theorem 1. Below we assume $\delta > 0$.

Only need to notice that $\left(S^{(t)}\right)^r$ is a union of open spheres with radius $r$ that are unconnected to each other. Using Lemma 2, we have $\left(S^{(t+1)}\right)^{(1-\eta K)r}$ is a union of open spheres with radius $(1 - \eta K)r$ that are unconnected to each other. Since any two open spheres are homeomorphic, we have $\left(S^{(t)}\right)^r$ is homeomorphic to $\left(S^{(t+1)}\right)^{(1-\eta K)r}$.

## B  EXTRA EXPERIMENT RESULTS

In this section, we provide extra experiment results performed with more optimizers, complementing those in Section 6.

**Low Dimensional Neural Networks**  We extend the low-dimensional experiments to additional optimizers. Figure 6 presents addition result for the 2D network trained with GD, under a different initialization. Figures 9 and 10 present the results for 2D and 3D networks trained with Adam, and Figures 7 and 8 present the corresponding results with momentum gradient descent.

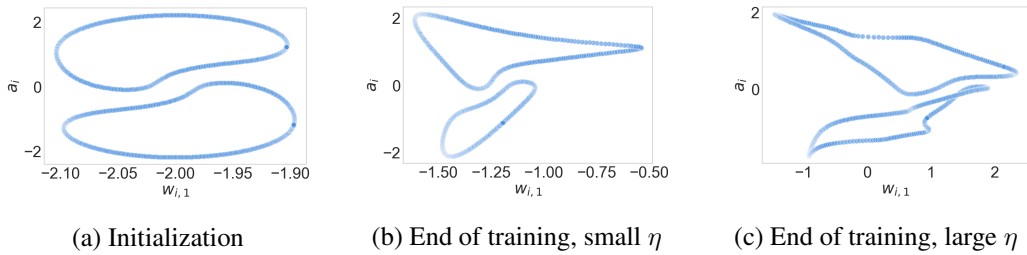

(a) Initialization     (b) End of training, small $\eta$     (c) End of training, large $\eta$

Figure 6: **Topology of a 2D neural network with GD and disjoint genus-1 initialization.** The neurons are initialized on the disjoint union of two genus-1 surfaces and optimized with GD.

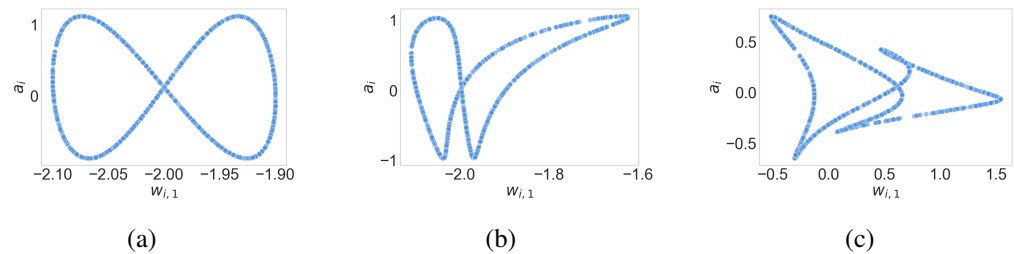

(a)          (b)          (c)

Figure 7: **Topology of a 2D neural network with momentum gradient descent.** The neurons are initialized on a genus-2 surface and optimized with momentum GD.

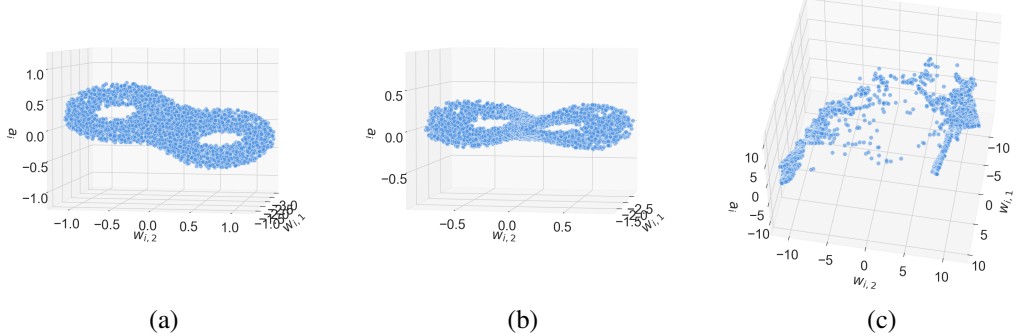

(a)          (b)          (c)

Figure 8: **Topology of a 3D neural network with momentum gradient descent.** The neurons are initialized on a genus-2 surface and optimized with momentum GD. The camera angle is manually adjusted to better visualize the structure of the point cloud.

**Extra Results Complementing the Experiments on Real Tasks**  The Betti number results of two-layer networks on MNIST are presented in Figure 11. Notice that in the small step-size setting of Figure 11, the topology remains unchanged initially but begins to change after a certain period of training. We attribute this to a key difference in the dynamics of Adam under small versus large

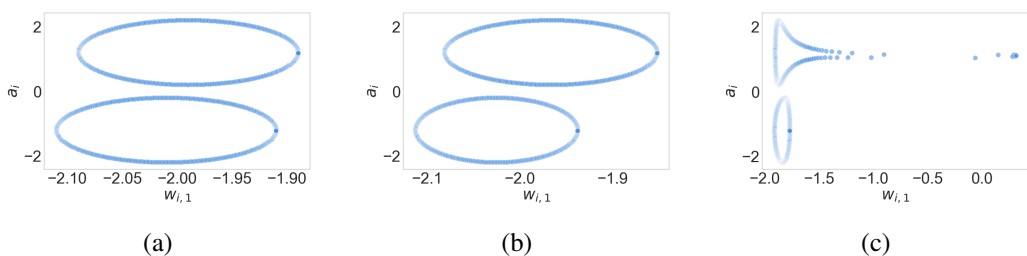

(a)                  (b)                  (c)

Figure 9: **Topology of a 2D neural network with Adam.** The neurons are initialized on the disjoint union of two genus-1 surfaces and optimized with Adam.

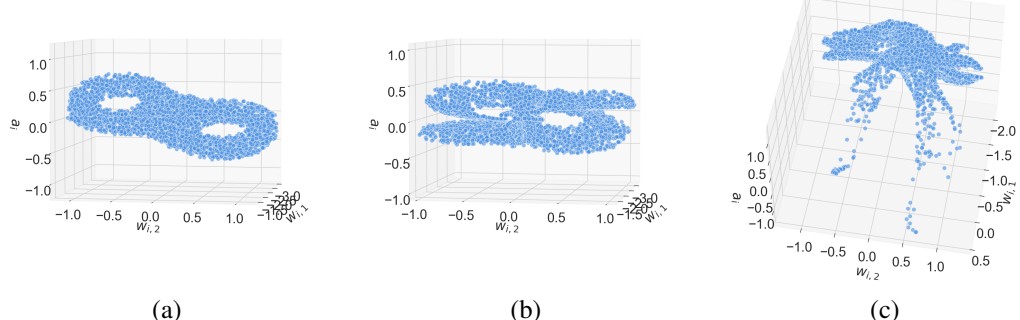

(a)                  (b)                  (c)

Figure 10: **Topology of a 3D neural network with Adam.** The neurons are initialized on a genus-2 surface and optimized with Adam. The camera angle is manually adjusted to better visualize the structure of the point cloud.

learning rates. Specifically, with small learning rates, the network undergoes progressive sharpening: the sharpness steadily increases and eventually surpasses the topological critical point. Beyond this point, the step size becomes relatively "large," and the topology of the neurons starts to change. In contrast, with large learning rates, the sharpness remains small. Similar phenomena have also been reported in the literature (Kalra & Barkeshli, 2023).

To verify this explanation, Figure 12 shows the evolution of the sharpness inversion ($1/K$, where $K$ denotes the largest eigenvalue of the Hessian matrix[5]) under Adam. Comparing these results with those in Figure 11, it is evident that the topology begins to change once $1/K$ becomes sufficiently small, matching our theoretical prediction.

**Extra Results with Multi-Layer Neural Networks** In order to further verify our theoretical results, we also conduct experiments with multi-layer neural networks. Specifically, we use a four-layer neural network, where the outputs of the first and third layers are wide (corresponding to the number of neurons), and the output of the second layer is 1-d (corresponding to the dimensionality of neurons). In this network, the weights in the first and second layer constitute a set of neurons, and so do the weights in the third and fourth layer, where the dimensionality of each neuron is 2. In Figure 13, we plot the initialization and end of training neuron distribution for both layers. Other than the model, the rest of the settings are the same as those in Appendix C.1. The results clearly support our theoretical claims.

**Extra Results with Loss of Plasticity** In order to further verify the effect of neuron topology simplification on the performance of the neural network, we conduct an extra experiment with multi-task training. Specifically, we first define a random task-generating process. In each epoch, we regenerate the data with a new random task and continue training the same model. See Appendix C.3 for the details of the setting. In Figure 14, we present the loss and the number of collapsed neuron

---

[5]The sharpness is calculated with the hessian-eigenthings library (Golmant et al., 2018).

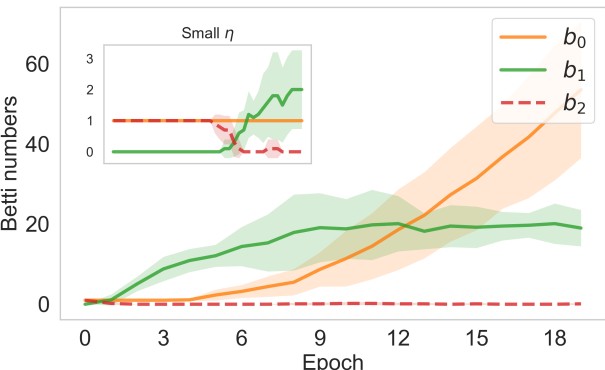

Figure 11: **Evolution of Betti numbers during training with Adam.** The plots show the first three Betti numbers $b_0$, $b_1$, and $b_2$ over time. The main panels correspond to large learning rates, while the insets show the results for small learning rates. Each curve is obtained by averaging over 10 runs with different random seeds; the curves denote the means and the shaded regions indicate the standard deviations. When the step size is small, the topology eventually changes after a certain training time. We attribute this to increasing sharpness: as training progresses, the network becomes sharper and the threshold for topological changes correspondingly decreases.

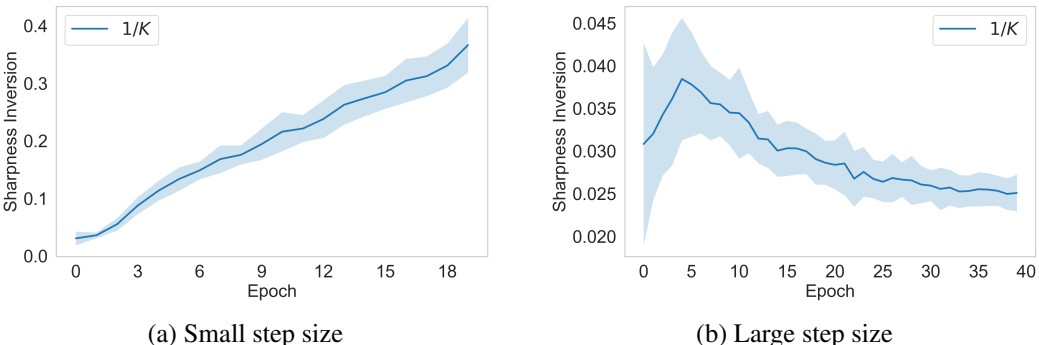

(a) Small step size        (b) Large step size

Figure 12: **Evolution of Betti numbers and sharpness inversion under Adam.** Here $K$ denotes the largest eigenvalue of the Hessian matrix. The small step-size setting is trained for a longer time to ensure convergence.

pairs after each epoch. It is clear that, in the large learning rate setting, the model is not able to learn, and the number of collapsed neurons is increasing, showing the loss of plasticity of the neural network; while in the small learning rate setting, there are no collapsed neurons and the model is able to continually learn the tasks even when they are changed at each epoch.

## C    EXPERIMENT DETAILS

In this section, we provide the experimental details.

### C.1    EXPERIMENTS WITH LOW-DIMENSIONAL NEURAL NETWORKS

As described in Section 6, we use a two-layer neural network with sigmoid activation, with input dimension $d = 1$ (referred to as the 2D case) or $d = 2$ (referred to as the 3D case).

Given input dimension $d$, the input data are denoted by $\mathcal{D} = \{(\boldsymbol{z}_s, y_s^*)\}_{s=1}^n \in \left(\mathbb{R}^d \times \mathbb{R}\right)^n$, where $n$ is the dataset size. Each input $\boldsymbol{z}_s$ is sampled from a Gaussian distribution with variance $4$, i.e., $\boldsymbol{z}_{s,j} \sim \mathcal{N}(0, 4)$ for $j \in \{1, 2\}$. The labels $y_s^*$ are generated by a teacher model

$$y_s^* = \langle \boldsymbol{a}^*, \sigma\left(\boldsymbol{W}^* \boldsymbol{z}_s\right) \rangle, \tag{59}$$

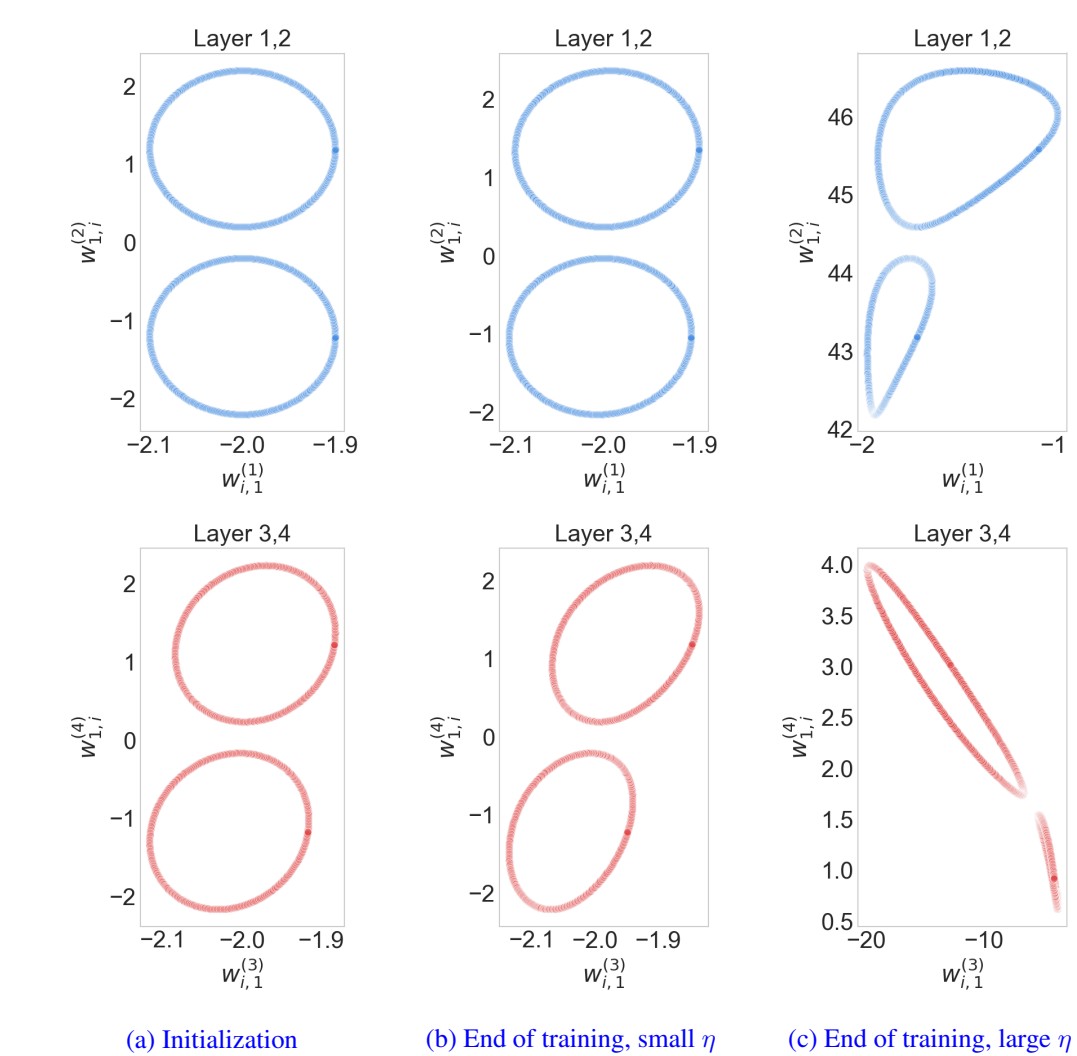

(a) Initialization  (b) End of training, small $\eta$  (c) End of training, large $\eta$

Figure 13: **Topology of a multi-layer 2D neural network with gradient descent.** The neurons are initialized on a genus-2 surface and optimized with GD.

where $\boldsymbol{a}^* \in \mathbb{R}^{h^*}$, $\boldsymbol{W}^* \in \mathbb{R}^{h^* \times d}$, and $h^*$ is the hidden size of the teacher model. Both $\boldsymbol{a}^*$ and $\boldsymbol{W}^*$ are randomly sampled at the beginning and fixed when constructing the dataset, with $a_j^* \sim \mathcal{N}(0,1)$ and $w_{j,k}^* \sim \mathcal{N}(0, 0.36)$. In all experiments, $n$ is set to 5000, with 70% of the data used for training. The model is trained using mini-batches of size 128. For GD with momentum, the momentum coefficient is set to 0.9.

Since different methods admit different thresholds for effective learning rates, we manually tuned the step sizes for each optimizer. The learning rates used to generate the reported results are summarized in Table 1. In all cases, we train the model until the training loss converges.[6]

### C.2 Experiments with Large Neural Networks

In the experiments on MNIST (Section 6), we use a two-layer MLP with sigmoid activation and hidden size 1024. The model is trained for classification using cross-entropy loss, without any

---

[6]Although in some cases the small and large step sizes appear close, we observed that low-dimensional networks are highly sensitive to the learning rate when trained with GD, possibly due to a degenerated loss landscape. For instance, in the 2D case, if $\eta = 2 \times 10^{-3}$ the neurons remain nearly unchanged, whereas for $\eta = 4 \times 10^{-3}$ the loss diverges. Thus we must compare within a relatively narrow range of learning rates.

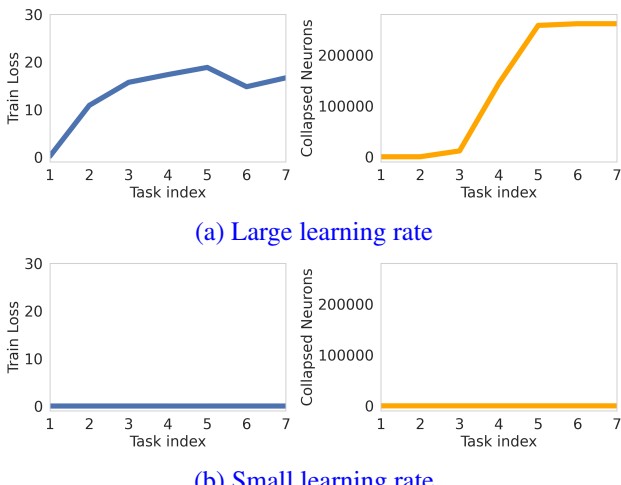

(a) Large learning rate

(b) Small learning rate

Figure 14: **The Multi-task learning results.** The neurons are initialized on a genus-2 surface and optimized with GD. After each epoch, the teacher model is resampled and the data is regenerated. "# of collapsed neurons" is calculated by counting the number of neuron pairs whose distance are smaller than $0.01$.

| Optimizer | Network dimension | Small learning rate | Large learning rate |
|-----------|-------------------|---------------------|---------------------|
| GD | 2D | $2.5 \times 10^{-3}$ | $3 \times 10^{3}$ |
| GD | 3D | $8 \times 10^{-4}$ | $9 \times 10^{-4}$ |
| Adam | 2D | $10^{-4}$ | $10^{-2}$ |
| Adam | 3D | $3 \times 10^{-2}$ | $10^{-1}$ |
| Momentum | 2D | $5 \times 10^{-4}$ | $4.5 \times 10^{-3}$ |
| Momentum | 3D | $10^{-3}$ | $1.25 \times 10^{-3}$ |
| Multi-Layer | N/A | 0.01 | 0.05 |

Table 1: Learning rates used in the experiments for low-dimensional neural networks.

additional regularization (e.g., weight decay). The batch size is set to $1024$. For GD, the small and large learning rates are $0.02$ and $0.5$, respectively. For Adam, the corresponding values are $10^{-5}$ and $10^{-3}$.

The Betti numbers are calculated with the GUDHI library (Maria et al., 2025). When computing Betti numbers for the neuron-induced point cloud, a minimal distance (i.e. scale) must be chosen to decide how close two points need to be for them to be considered as neighbors. To ensure robustness to scale changes during training, we adopt a self-adaptive strategy for deciding the minimal distance: the minimal distance is set to $1/4$ times the diameter of the point cloud.

## C.3 EXPERIMENTS WITH LOSS OF PLASTICITY

In the experiment with multi-task training, the input data are i.i.d. standard Gaussian vectors with dimensionality $8$. For each task, we sample a random linear mapping from the input data to a 8-dimensional space, as the teacher model, that acts on the input data to generate the target data. The input data and teacher model are resampled every epoch to change the task. Each task contains $50000$ training samples and $10000$ evaluation samples. The model is trained with GD on the mean-square-error (MSE) loss. The model structure is the same as that in Appendix C.2, with random initialization. For large and small learning rate settings, the learning rates are set to $0.25$ and $0.01$, respectively, and the weight decay rate is set to $0.01$. The first task (with index 0) is viewed as a warp-up and discarded.

# D  A Weaker Version of Continuity Property

In Section 3, we mentioned that the specific form of $K$-continuity property is only for establishing a correspondence with the smoothness property used in optimization theory, and in our theory this property can actually be weaker. Specifically, when $I$ is an infinite set, the $K$-continuity property defined in Section 3 implicitly requires that $U(\mathcal{X})$ and $U(\mathcal{Y})$ only differ in finite number of entries, which might not always hold. This condition can be relaxed to the upper bound on each entry of $U(\mathcal{X}) - U(\mathcal{Y})$.

**(P2′) Altered $K$-Continuity Property**: If there exists a constant $K > 0$, such that for any $t \in \mathbb{N}$, $\mathcal{X}, \mathcal{Y} \in \left( \mathbb{R}^D \right)^I$ and $i \in I$, we have

$$\sup_{i \in I} \left\| U_i^{(t)}(\mathcal{X}) - U_i^{(t)}(\mathcal{Y}) \right\| \le \frac{K}{2} \left\| \mathcal{X} - \mathcal{Y} \right\|, \tag{60}$$

then we say $U^{(t)}$ has altered $K$-continuity property.

The proof of the main theories is built upon Lemma 4. Here we prove a variation of Lemma 4 with the altered $K$-continuity property.

**Lemma 5** (Altered - No Splitting). *The following statement holds when $U$ has the equivariance property and altered $K$-continuity property. For any $t \in \mathbb{N}$ and $i, j \in I$ such that $i \ne j$, we have*

$$\left\| U_i(\mathcal{X}^{(t)}) - U_j(\mathcal{X}^{(t)}) \right\| \le K \left\| \boldsymbol{x}_i^{(t)} - \boldsymbol{x}_j^{(t)} \right\|. \tag{61}$$

*Proof.* Let $\mathcal{X} = \mathcal{X}^{(t)}$ for convenience. Define $P \in \mathsf{FSym}(I)$ as switching $i$ and $j$ (as in eq. (20)). Then using the equivariance property, we have

$$U_i(\mathcal{X}) - U_j(\mathcal{X}) = U_i(\mathcal{X}) - \left( PU(\mathcal{X}) \right)_i = U_i(\mathcal{X}) - U_i(P\mathcal{X}). \tag{62}$$

Notice that $\mathcal{X}$ and $P\mathcal{X}$ only differ in entries $i$ and $j$. Using the $K$-continuity property, we have

$$\left\| U_i(\mathcal{X}) - U_j(\mathcal{X}) \right\| = \left\| U_i(\mathcal{X}) - U_i(P\mathcal{X}) \right\| \tag{63}$$

$$\le \frac{K}{2} \left\| \mathcal{X} - P\mathcal{X} \right\| \tag{64}$$

$$\le K \left\| \boldsymbol{x}_i^{(t)} - \boldsymbol{x}_j^{(t)} \right\|. \tag{65}$$

$\square$

In all the presented theories, the $K$-continuity property can be replaced by the altered $K$-continuity property. The proofs directly apply by replacing Lemma 4 with Lemma 5.

## Large Language Model Usage

In preparing this submission, we used a large language model (ChatGPT) as an assistive tool for language polishing. The model did not contribute to the research ideation, experimental design, data analysis, or the generation of scientific content. All substantive content, results, and conclusions presented in this paper were conceived, written, and verified by the authors, who take full responsibility for the work.

