# OpenReview forum: "Topological Invariance and Breakdown in Learning"
_ICLR.cc/2026/Conference — Submitted to ICLR 2026_

### Official Review · Reviewer_9kDK · 2025-10-30

**Soundness:** 2
**Presentation:** 3
**Contribution:** 2
**Rating:** 2
**Confidence:** 4

**Summary:**

Permutation symmetry allows us to view the neurons inside a neural network as points belonging to a set that we can interpret as a topological space. Permutation symmetry and "continuity" of the training algorithm can be used to prove that each training step is a homeomorphism (a topology-preserving map) for all learning rates lower than a certain *topological* threshold. When the learning rate is higher than this threshold, the training algorithm can collapse different neurons and change the neurons' topological structure, simplifying it. The results also show that, below the threshold, the probability density of neurons is also preserved. Simple numerical experiments on two-layer neural networks validate this when the neurons are initialized on toy manifolds.

**Strengths:**

- I think the approach proposed in the paper is novel and promising. The idea that there is a topological threshold below which topology of the neurons, in some sense, preserved is quite interesting.
- The paper employs topology, an underexplored tool in the neural network literature, to study the joint effect of symmetry and finite learning rates, a significant and well-studied problem in current research.
- The paper is well explained and clearly readable, even in its more technical aspects.

**Weaknesses:**

As I wrote above, I think the ideas presented in the paper are quite interesting. However, I see two main issues with the theoretical contributions that, at the moment, prevent me from recommending acceptance.

The main weakness of the paper, in my view, is that in all real scenarios, the number of neurons is, of course, finite. This means that the set of neurons $S^{(t)}$ will be a discrete set of different (with prob 1 if we initialize randomly) points in Euclidean space. If we make $S^{(t)}$ inherit the Euclidean topology of $\mathbb{R}^d$ (as prescribed in L222), it will inherit the *discrete topology*, i.e. the "finest" possible one generated by all subsets of $S^{(t)}$.
Therefore, in all real cases where neurons are neither continuous nor infinite, the neurons set can't be as structured as a topological or differentiable manifold.

The main result thus reduces to saying that, below the topological threshold, the training algorithm will never merge two points into the same one. This comes from the fact that any bijective function between spaces with the discrete topology is a homeomorphism, as any function will be continuous. Put in this way, I think, the result is not particularly interesting or surprising as we are talking about points into a continuous space and the fact that two of them will exactly coincide is, intuitively, a "measure zero event" which will therefore virtually never happen.

The result would be more interesting if the authors were able to prove that a "coarser" notion of topology, intended in the sense of topological data analysis, is preserved. This would mean intuitively that we see the neuron as "approximating" a topological space or a manifold and the topological features are defined, e.g. by persistent homology. The authors already seems to suggest that this is the case in the numerical experiments, but I don't think that the theoretical result implies it in any way.

Moreover, usually weights are randomly sampled, and it seems unlikely that neurons will fall in such a pattern that can resemble the approximation of a manifold, like a Gaussian blob. It is not clear what preservation/non-preservation of topology would entail in that scenario.

The second, somewhat less important issue I see, is that in a standard MLP neural network, only neurons belonging to the same layer can be swapped, that is permutation symmetry only holds layerwise and the result of the paper only hold in each layer separately. I believe that this further reduces the impact of this set of results.

### Minor weaknesses
- As I mentioned above, the main topology-preservation result tells us that neurons points cannot merge (as it is also recognized in the paper at line L245). This is discussed in *Simsek et al. "Geometry of the loss landscape in overparameterized neural networks: Symmetries and invariances." International Conference on Machine Learning. PMLR, 2021*, where it is proven that gradient flow cannot reach a symmetry subspace (equivalent to merging neurons states) in finite time. I recognized that the authors deal with a more general setting, but I think this should be acknowledged.
- L097: shouldn't "row" and "column" be swapped?
- L119: "the collection of neurons at a given time step can be viewed as a manifold embedded in the Euclidean space, whose topology reflects its connectivity and shape." I don't think this is a correct characterization of the works referenced after this sentence. They each study the topology of neural networks in different senses, all of which are different from the one explored in this work.

**Questions:**

- Referring to my comments above, do the authors believe that their results extend to a notion of topology besides the "trivial" homeomorphism between discrete finite spaces, like it is suggested in the experiments?
- If this is the case, how can we interpret the result when we initialize weights randomly and not on a well-behaved manifold?

---

> ### Author Response · Authors · 2025-11-24
> **Response to Reviewer 9kDK (Part 1)**
>
> We thank the reviewer for recognizing the novelty and promise of our approach, as well as the clarity of the writing. We understand that the reviewer's main concerns focus on two aspects: (1) the practical relevance of the proposed perspective, and (2) the scope and applicability of the permutation symmetry. We address each of these points in detail below.
>
> #### **W1. Regarding practical relevance**
>
> We would like to emphasize that the core idea of this paper is straightforward: when the learning rate is small, the update map cannot substantially alter the structure of the neurons and remains invertible; when the learning rate is large, the update map can induce non-invertible changes. Topology serves mainly as a precise language for expressing this dichotomy. The topological viewpoint is particularly natural when one interprets the neuron set in the infinite-width limit, which is consistent with many theoretical works adopting infinite-width approximations [1, 2]. A similar "neural manifold" perspective has also been widely used in neuroscience [3].
>
> That said, even under the discrete topology of a finite neuron set, our results still yield nontrivial predictions: if two neurons collapse to the same point, they cannot be separated again under the learning algorithm (continuity); and when the learning rate is small, such collapses cannot occur in the first place (homeomorphism). Although simple to state, these facts are far from trivial. For example, consider the function $f(x,y) = ax + by$ with $a \neq b$. Even when $x = y$, a gradient-based update can separate the two coordinates because their gradients differ. The contuity of gradient methods only holds in the presence of permutation symmetry, which is precisely what our framework captures.
>
> The reviewer also pointed out that randomly initialized weights need not form a manifold. While this is correct, our theory does not need to be applied strictly from initialization. In later stages of training (or during fine-tuning), it is plausible that neurons already lie close to a low-dimensional structure. Questions such as whether a model can acquire new knowledge or unlearn previously acquired knowledge naturally relate to the topology of the neuron configuration, and we believe our theory is relevant in this context.
>
> We hope these clarifications address the reviewer's concerns. For a more comprehensive discussion, please refer to our General Response.
>
> #### **W2. Regarding the scope of permutation symmetry**
>
> Your understanding is correct: in a standard MLP, permutation symmetry operates layerwise, and therefore our results apply to each layer separately. We believe this is a natural and appropriate level of granularity for analysis. Neurons in different layers typically serve different computational roles and may even have different dimensionalities, making it inherently less meaningful to compare or permute them across layers. Analyzing layerwise permutation symmetry is thus consistent with both the architecture and the standard theoretical treatment of neural networks.
>
> ### Minor Weaknesses
>
> #### **Regarding the suggested reference.**
>
> We thank the reviewer for suggesting this reference. We will add it to the background section in the revision.
>
> #### **L097: shouldn't "row" and "column" be swapped?**
>
> You are right, this is a typo. We have fixed it in the revision.
>
> #### **Regarding L119.**
>
> Thank you for pointing this out. We have revised the statement to provide a clearer description of the cited work.
>
>
> ### Questions
>
> #### **Q1. Do the authors believe that their results extend to a notion of topology beyond the "trivial" homeomorphism between discrete finite spaces, as suggested in the experiments?**
>
> Please see our explanation above. We believe that topology is the most suitable language for expressing our idea, provided we interpret the neurons as approximately forming a manifold in the large-width regime.
>
> #### **Q2. If this is the case, how can we interpret the result when we initialize weights randomly and not on a well-behaved manifold?**
>
> Please see our earlier response under "practical relevance," where we discuss how the theory applies even when initialization does not lie on a well-structured manifold.
>
> ### References
>
> [1] Tensor Programs I: Wide Feedforward or Recurrent Neural Networks of Any Architecture are Gaussian Processes, NeurIPS 2019
>
> [2] Mean-field theory of two-layer neural networks: dimension-free bounds and kernel limit, COLT 2019
>
> [3] A neural manifold view of the brain, Nature Neuroscience 2025

---

> > ### Comment · Reviewer_9kDK · 2025-11-26
> >
> > I thank the authors for their response, the updates to the document and the extra experiments.
> >
> > Overall, the paper idea is quite interesting, but I still think that the current framing is not adequately motivated and could potentially mislead the reader into overestimating the applicability of the main theorem.
> >
> > Let me explain once again my concern. The central result only shows the preservation of the topology in the *strict sense*, that is, the one induced by the Euclidean topology on the neuron set.
> > This means that, when the neurons are finite (even if they approximate a manifold), the theorem only tells us that the states of two different neurons cannot collapse to the *exact same* value.
> > This may be non-trivial, like the authors write in the general response, but is far from the statement the topology of the manifold/topological space these neurons approximate is preserved.
> >
> > In fact, even the plots in Figs. 3-4 for large learning rates display homeomorphisms in the "strict" sense (provided that there are no two points which are *exactly* equal, which seems to me to be the case).
> >
> > The authors write that *"[...] a common approach in theoretical deep learning is to analyze learning dynamics in the large-width (infinite-width) limit"*.
> >
> > This is true as infinite-limit analysis provided many insights into the training dynamics. However, I think that, even if we assume that the neurons are infinite, the fact that *"the neuron set can be viewed as a discretization of an underlying continuous manifold"* (at initialization or in later stages of training) seems to me to be unjustified and would need to be experimentally or theoretically shown. Having an infinite number of neurons and they having the structure required for them to be a manifold are two different things, the second being a much stronger assumption.
> > The paper cited indeed study infinite width/depth limits. However, I don't see how taking this limit, which gives an infinite but *countable* set of neurons, would result in an approximation of a manifold.
> > Moreover, as I wrote above, even if they did approximate a manifold, the main theorem wouldn't tell us anything about the preservation of the topology of the manifold they approximate.
> >
> > I still want to stress that the paper's idea is interesting and could potentially be impactful if the main result was extended to account for a "softer" notion of topology, like the experiment suggest.
> > For instance, showing that the result extends to the topological spaces obtained by taking the union of the balls of a certain radius centered on the neurons, would give it much more significance.

---

> ### Author Response · Authors · 2025-11-26
> **Reply**
>
> Thank you for the additional comments and for the careful reading of our response. We appreciate your suggestion on extending the theoretical results to a "softer" notion of topology. In response to this, we have now added an additional theorem (Theorem 2) in the revision. Theorem 2 defines a topology induced by discrete sets that allows one to study the overlap of nontrivial neighborhoods around neurons. Under this new topology, topological simplication directly means that two neurons are close within a finite radius. These open sets are equipped with a finite measure in the real space, and so any events of collapsing are no longer a probability-zero event. This addresses your primary concerns above. This provides a strictly stronger characterization of the neuron structure in Theorem 2, where simplification is no longer confined to exact equality between points but reflects a genuinely practically-relevant effect. This new result should also be extendable and/or generalizable to more interesting approximate topologies, which we leave as future work.
>
> Moreover, we would like to further clarify several points.
> 1. When the number of neurons is large, it is standard to interpret the population of neurons as sampling an underlying continuous object, as done in mean-field theory, NTK analyses, tensor programs, and many other theoretical frameworks. For example, in Figure 3, the figures were produced with 1024 neurons, and if one interpolates between the neurons shown, genuine topological simplification does occur in the large learning rate case, as the interpolated curve develops more than two holes, whereas the initialization has exactly two. We believe this shows that the infinite-width interpretation is indeed meaningful and the results in Figure 3 and 4 are consistent with our theory.
> 2. Even if one insists on considering only the discrete perspective, neuron merging is not a vacuous phenomenon. In the Loss of Plasticity (LoP) experiment added during the rebuttal stage (summary provided in the General Response), we explicitly measured and reported the number of merged neurons under large learning rates. The results show extensive neuron merging happening. Thus, even if one insists on interpreting the theory strictly in the discrete topology, the phenomenon of collapse is real, observable, and practically relevant.
>
> We thank you again for your thoughtful feedback and constructive suggestions. We hope that the above clarifications, together with the additional results and experiments included in the revised manuscript, satisfactorily address your concerns and convey the intended scope and significance of our work.

---

### Official Review · Reviewer_Ws8X · 2025-10-30

**Soundness:** 1
**Presentation:** 3
**Contribution:** 2
**Rating:** 2
**Confidence:** 3

**Summary:**

The paper studies how permutation-equivariant learning rules (e.g., SGD, Adam) affect the topological structure of neurons during training. The authors prove that for sufficiently small learning rates (below a critical threshold $\eta^*$), training acts as a bi-Lipschitz mapping and preserves the topological features of the neuron manifold. Once the learning rate exceeds this threshold, training enables topological simplification: neuron manifolds become coarser, expressive capacity may reduce, and the model structure simplifies. The proposed theory is claimed to be architecture- and loss-function-agnostic, offering a general framework for understanding the interplay between learning rate and the topology of neural representations.

**Strengths:**

The idea of analyzing training dynamics through the topology of neuron configurations is original and conceptually appealing. Connecting learning-rate regimes with structural changes in neural representations could offer new perspectives on optimization stability and expressivity.

**Weaknesses:**

While the framework is intriguing, its topological interpretation remains unclear. In practical settings, the number of neurons is finite, so the natural topology on this set is discrete, where every bijection is trivially a homeomorphism. Intuitively, small learning rates cause small parameter updates, so the point cloud of neurons should change smoothly. However, it is not evident in what sense nontrivial topological properties arise or how they meaningfully relate to the learning dynamics.

The paper further suggests that a coarser topology corresponds to lower expressivity and that stable optimization requires topological preservation. These claims are intriguing but not well justified. Clarification on what ``preserving topology'' means in the finite or discrete setting, and how this affects training dynamics and expressivity, is crucial to assess the paper’s relevance.

**Questions:**

- Which topological properties are considered, and how do they relate to training dynamics?
- Why does a coarser topology of the neuron manifold imply lower expressivity? How does the topology of weights constrain the representable function class?
- Why should stable optimization require the updating map to be a homeomorphism (or bijection in the discrete case)?
- For a finite neuron set, one could induce nontrivial topologies using tools from topological data analysis (e.g., Vietoris--Rips complexes). Could the authors comment on why they did not adopt such constructions and instead worked with the discrete topology?

---

> ### Author Response · Authors · 2025-11-24
> **Response to Reviewer Ws8X (Part 1)**
>
> We thank the reviewer for recognizing the originality and potential impact of our work, and for providing constructive feedback. The reviewer raises two central concerns: 1) how our topological framework connects to realistic finite-width neural networks, and 2) whether the claim that coarser topology implies reduced expressivity is sufficiently supported. We address each of these points in detail below.
>
> #### **it is unclear how the topological perspective is connected with realistic settings where the number of neurons is finite**
>
>
> We thank the reviewer for raising this important point. We would like to clarify how the topological perspective connects to realistic finite-width neural networks. The central idea of our work is that small learning rates lead to invertible evolution of neurons (no collapse), whereas sufficiently large learning rates can induce non-invertible changes. Topology provides a natural and mathematically precise language to formalize this distinction. For finite neuron sets, this perspective can be interpreted in two complementary ways:
> - **Large-width (infinite-width) viewpoint.** When the number of neurons is large (which is typical in modern networks), the discrete set can be viewed as a discretization of an underlying continuous manifold. This is standard in many theoretical treatments of wide networks, such as [3, 4].
> - **Finite-set (discrete topology) viewpoint.** For a finite discrete set, continuity means that neurons cannot be split apart once they collapse, and homeomorphism means that neurons do not collapse in the first place. Notice that these conclusions are not trivial as they are a consequency of the permutation symmmetry of the neuron networks.
>
> In summary, whether interpreted via a large-width manifold limit or via the discrete topology of finite neurons, the topological perspective provides meaningful and nontrivial insights into realistic training dynamics. For a more comprehensive discussion, please refer to our general response.
>
> #### **the claim of coarser topology corresponds to lower expressibity is not well-supported.**
>
> The link between topological simplification and reduced expressivity arises naturally from the structural constraints imposed by the learning dynamics.
>
> Firstly, it is known that topological simplification can not be reverted by continuous mappings. Our theoretical result shows that the learning dynamics are continuous, thus once neurons collapse (i.e. becomes sufficiently close) and the topology becomes simpler, the optimization trajectory cannot revert to a topologically richer configuration. Since different topological structures correspond to function families that cannot be continuously deformed into one another, this imposes an inherent structural restriction on the model's expressivity. In other words, the model becomes incapable of representing functions associated with the lost topological degrees of freedom.
>
> Moreover, To provide additional empirical support, we conducted an experiment showing that large learning rates lead to loss of plasticity [1], where the model becomes unable to acquire new tasks in a continual learning setting. This behavior aligns with the theoretical prediction that topology coarsening restricts the reachable solution space. See Appendix B and Figure 14 for the corresponding results (new content added during the revision stage is marked in blue).
>
> Together, these theoretical and empirical observations support the interpretation that topological simplification corresponds to reduced expressivity.

---

> ### Author Response · Authors · 2025-11-24
> **Response to Reviewer Ws8X (Part 2)**
>
> ### Questions
>
> Below we answer the questions raised by the reviewer.
>
> #### **Q1. Which topological properties are considered, and how do they relate to training dynamics?**
>
> We consider the topology of the neuron manifold as a whole, rather than individual topological properties. Since a homeomorphism preserves all topological properties, our theoretical statements concern whether the learning dynamics induce homeomorphic or non-homeomorphic transformations of the neuron manifold. This distinction captures exactly when the structural configuration of neurons is preserved or simplified during training.
>
> #### **Q2. Why does a coarser topology of the neuron manifold imply lower expressivity? How does the topology of weights constrain the representable function class?**
>
> Under the discrete viewpoint, a topological simplification corresponds to neurons collapsing into a single point. Our theory shows that such collapses are irreversible under continuous learning dynamics, so the model effectively loses degrees of freedom, as it can no longer separate those neurons in future training steps. This reduction in degrees of freedom naturally leads to a reduction in expressivity, as fewer distinct neuron parameters are available to represent diverse functions.
>
> A similar idea appears in prior work showing that merging neurons reduces the representational capacity of neural networks (e.g., [2]). Please also see our detailed response to the weaknesses above for further discussion.
>
> #### **Q3. Why should stable optimization require the updating map to be a homeomorphism (or bijection in the discrete case)?**
>
> To clarify, our paper does not claim that "stable optimization requires a homeomorphism." The term stable can be ambiguous: practitioners sometimes describe training as "stable" whenever the loss is decreasing, even though it might not be stable in the classical sense of convex optimization. In our paper, we use "stable" in the strict sense from optimization theory, where small perturbations in parameters lead to small perturbations in updates. Under this interpretation, the update map becomes continuous and injective, which is why homeomorphism naturally arises in our characterization. But we do not assert that this is the only notion of stability.
>
> #### **Q4. For a finite neuron set, one could induce nontrivial topologies using topological data analysis (e.g., Vietoris–Rips complexes). Why not adopt such constructions instead of the discrete topology?**
>
> We appreciate this suggestion. In fact, for the experimental part of the paper (e.g., when computing Betti numbers), we do use Rips complexes to obtain meaningful topological summaries of the point clouds formed by neuron embeddings.
>
> regarding the theoretical results, we believe that standard topology provides the most appropriate and natural language to express our ideas. It is also important to clarify that we conceptually view the neurons as forming an infinite set and thus do not rely on the discrete topology.
>
> ### References
>
>
> [1] Loss of plasticity in deep continual learning, Nature 2024
>
> [2] A Regularity Condition of the Information Matrix of a Multilayer Perceptron Network, Neural networks 1996
>
> [3] Tensor Programs I: Wide Feedforward or Recurrent Neural Networks of Any Architecture are Gaussian Processes, NeurIPS 2019
>
> [4] Mean-field theory of two-layer neural networks: dimension-free bounds and kernel limit, COLT 2019

---

### Official Review · Reviewer_6rk3 · 2025-10-30

**Soundness:** 2
**Presentation:** 2
**Contribution:** 2
**Rating:** 4
**Confidence:** 2

**Summary:**

The paper studies a class of permutation-equivariant learning rules (including SGD, Adam, and others).
Authors show that training process induces a bi-Lipschitz mapping of neurons and preserves key topological properties of the neuron distribution. This result reveals a qualitative difference between small and large learning rates.

**Strengths:**

1. The paper theoretically studies a "topology" of sets of neurons. Authors prove that "topology" changes smoothly for small learning rates
2. The theory is universal across architectures (MLP, CNN)
3. The paper grounds its claims in well-established concepts from topology (homeomorphisms, quotient maps, Betti numbers) and measure theory, offering a formal language to describe neural network evolution.

**Weaknesses:**

1. The situation when neurons (weights) lie on a genus-2 manifold is artificial. Typically, neurons are initialized randomly.
2. Experiment are limited to small MLPs and MNIST. Large models, like Transformers are not studied.
3. I am not sure that the definition of "neuron" and equivariance (eq. (2)) are applicable for transformers.
4. The generalization error, which is of most interest in ML/AI is not studied.
5. Despite of the theoretical insights, the paper doesn’t propose new training algorithms, regularizers, or learning rate schedules that exploit the topological critical point.

**Questions:**

1) In Equation 4, instead of X must be X^{t} ?
2) How your paper is related to:
Naitzat, G., Zhitnikov, A., & Lim, L. H. (2020). Topology of deep neural networks. Journal of Machine Learning Research, 21(184), 1-40.
3) Fig. 3, 4. When you write "neurons are initialized on genus-2 surface" it is not clear what you mean by "neurons".
You mean activations of weights of a network? Because "X" in previous equations mean weights, for which evolution rule (4) applies.
4) How discrete set of neurons can evolve by a homeomorphism?
5) How your results are related to Edge of Stability (EOS) ?
Zhu, X., Wang, Z., Wang, X., Zhou, M., & Ge, R. (2022). Understanding edge-of-stability training dynamics with a minimalist example. arXiv preprint arXiv:2210.03294.
6) Can we compute or estimate K empirically?

---

> ### Author Response · Authors · 2025-11-24
> **Response to Reviewer 6rk3 (Part 1)**
>
> We thank the reviewer for recognizing that our theoretical result is universal and well-established. Below, we address each concern raised by the reviewer.
>
>
> #### **W1. The situation when neurons (weights) lie on a genus-2 manifold is artificial. Typically, neurons are initialized randomly.**
>
> We appreciate the reviewer's concern and would like to clarify that the use of a genus-2 manifold in some experiments is purely for illustrative purposes, to make topological changes visually interpretable. Our theoretical results do not rely on this choice in any way.
>
> For standard random initializations (uniform or Gaussian), when the network is wide enough, the induced neuron distribution is approximately homeomorphic to a hypercube, and the corresponding topological changes typically manifest as folding or stretching of this high-dimensional set. Such phenomena are considerably harder to visualize directly, which is why a genus-2 manifold provides a clearer pedagogical example.
>
> Moreover, while current initialization schemes are predominantly based on such random distributions, a general theoretical framework should accommodate broader initialization geometries. Our analysis may thus inform or inspire future work on manifold-structured initialization strategies.
>
> Finally, prior work has shown that, during training, neuron embeddings often collapse onto lower-dimensional structures due to simplicity bias. This means that analyzing non-full-dimensional manifolds is not purely artificial; our experiments are directly relevant to the regimes after such collapses occur.
>
> #### **W2. Experiment are limited to small MLPs and MNIST. Large models, like Transformers are not studied.**
>
> Thank you for raising this point. As we will discuss below in the response to W3, the theoretical framework applies to any model architecture that contains components of the form $W_2\sigma(W_1 x)$. This structure appears not only in standard MLPs but also in the feed-forward layers of Transformers, as well as in the QK transformations used within the attention mechanism. Therefore, the theory is not restricted to small or specific networks and is directly applicable to widely used large-scale architectures. The experiments in the paper serve as empirical illustrations of the theoretical claims. To make the topological evolution easier to visualize and interpret, we chose small models and datasets, where the geometry can be inspected more transparently. Overall, the theoretical results are architecture-agnostic, while the experimental choices are driven by clarity rather than limitations of the theory.
>
> #### **W3. I am not sure that the definition of "neuron" and equivariance (eq. (2)) are applicable for transformers.**
>
> Thank you for raising this point. We clarify here why the definitions of "neurons" and the equivariance condition in Eq. (2) apply naturally to Transformer architectures.
>
> In the feed-forward layers of a Transformer, the mapping has the form $$x \mapsto W_2 \sigma(W_1 x),$$
> which matches exactly the bilayer structure considered in our theory, as stated in Section 2 of the paper. The concatenation of the $k$-th column of $W_2$ and $k$-th row of $W_1$, for each $k$, forms a neuron in our definition, and permutations of these units leave the overall function form invariant, satisfying the equivariance relation in Eq. (2).
>
> Furthermore, in the self-attention mechanism, the attention map is calculated by $$X \mapsto \mathop{ \mathrm{ softmax } }\left(X W_Q W_K ^\top X^\top \right),$$
> where $X \in \mathbb R^{n \times d}$ is the data matrix. In this case, $W_Q$ and $W_K^\top$ play the role of $W_2$ and $W_1$ above, respectively, so the pairing of the $k$-th column of $W_Q$ and $W_k$ forms a neuron, for each $k$.
>
> Thus, both the feed-forward and self-attention components of Transformers fall within the scope of the theoretical framework. We have revised the manuscript to make this point clearer. We have updated the manuscript to detail the concept of neurons in transformer models.

---

> ### Author Response · Authors · 2025-11-24
> **Response to Reviewer 6rk3 (Part 2)**
>
> #### **W4. The generalization error, which is of most interest in ML/AI is not studied.**
>
> We appreciate the reviewer's comment. Our work focuses on characterizing a fundamental aspect of learning dynamics, namely the topological evolution of neuron manifolds, rather than providing explicit generalization bounds. The primary goal is to identify structural constraints on the optimization trajectory that influence training phenomena such as the edge of stability [1] and loss of plasticity [2]. These behaviors are known to correlate with generalization quality, even though theoretical generalization guarantees remain an open challenge.
>
>
> That said, our results do have conceptual implications for generalization. Several recent works have demonstrated that topological or geometric properties of the learning dynamics can constrain or predict generalization performance (e.g., via Hausdorff dimension or heavy-tail behavior [3]). Since our framework characterizes how optimization preserves or alters topological structure, it provides a mechanism-level perspective that could serve as a foundation for topological approaches to generalization in future work. We have provided a General Response where we  discuss the potential impact of our theory in more details.
>
>
> #### **W5. Despite of the theoretical insights, the paper doesn’t propose new training algorithms, regularizers, or learning rate schedules that exploit the topological critical point.***
>
> We appreciate the reviewer's comment. We would like to clarify that the primary goal of this paper is to establish a theoretical characterization of how the topology of neuron manifolds evolves under gradient-based learning. As a theory-focused contribution, the work does not aim to propose new training algorithms or schedules directly. Instead, it provides a mechanism-level understanding that may inform future developments.
>
> That said, our framework does suggest several directions where practical methods could potentially be inspired:
>
> - **Generalization-related insights.** Our results show that learning dynamics impose topological constraints on the space of reachable solutions. Since existing work has linked geometric and topological properties to generalization behavior, our theory may inform future approaches that incorporate such constraints.
> - **Learning rate scheduling.**  Because topological transitions occur when the learning rate crosses a critical point, the topological viewpoint provides a principled way to interpret how schedules move the system between different dynamical regimes. This perspective may inspire new scheduling strategies in future work.
> - **Regularization.** Many regularization techniques influence sharpness, and our theory links sharpness directly to topological structure. This connection suggests that a topology-aware interpretation or new regularization principles could be developed from this perspective.
>
> In summary, while proposing new algorithms is not the aim of this paper, we believe that establishing a clear theoretical mechanism is valuable in its own right and can serve as a foundation for future method development. Please refer to the General Response for a more detailed discussion about the practical relevance of our theory.

---

> ### Author Response · Authors · 2025-11-24
> **Response to Reviewer 6rk3 (Part 3)**
>
> ### Questions
>
> Below we answer the questions raised by the reviewer.
>
> #### **Q1. In Equation 4, instead of X must be X^{t} ?**
>
> Thank you for pointing this out, and this is indeed a typo. We have fixed this in the revision.
>
> #### **Q2. How your paper is related to: Naitzat, G., Zhitnikov, A., & Lim, L. H. (2020). Topology of deep neural networks. Journal of Machine Learning Research, 21(184), 1-40.**
>
> This is a good question. The two works study fundamentally different objects: Our paper analyzes the topology of the neurons (i.e., trainable parameters) and how this topology evolves under the learning dynamics, while Naitzat et al. (2020) analyze the topology of hidden representations as they change along the forward pass. Thus, although both papers involve topological tools, the focus, mathematical objects, and goals are distinct. Moreover, our work is theory-driven and provides provable guarantees, whereas the referenced work is primarily empirical.
>
> #### **Q3. Fig. 3, 4. When you write "neurons are initialized on genus-2 surface" it is not clear what you mean by "neurons". You mean activations of weights of a network? Because "X" in previous equations mean weights, for which evolution rule (4) applies.**
>
> The definition of a "neuron" is explained in Section 2. Throughout Sections 3 and 4, the variable  $X$ consistently refers to the weights, and the evolution rule in Eq. (4) applies to these parameters. When we say "neurons are initialized on a genus-2 surface," we mean that the weight vectors (i.e., the parameterized neurons defined in Section 2) are initialized to lie on that manifold.
>
>
>
> #### **Q4. How can a discrete set of neurons evolve by a homeomorphism?**
>
> This is a good question. There are two complementary viewpoints to understand this:
>
> - **Large-width (infinite-width) viewpoint.** When the number of neurons is large (which is typical in modern networks), the discrete set can be viewed as a discretization of an underlying continuous manifold. This is standard in many theoretical treatments of wide networks, such as [4, 5].
> - **Finite-set (discrete topology) viewpoint.** For a finite discrete set, continuity means that neurons cannot be split apart once they collapse, and homeomorphism means that neurons do not collapse in the first place. Notice that these conclusions are not trivial as they are a consequency of the permutation symmmetry of the neuron networks.
>
>
> For a more comprehensive discussion, please refer to our General Response. Note that the finite-set viewpoint is particularly relevant in practice, and as demonstrated in our new experiment, finite-width systems can indeed exhibit observable topological collapses. We have also included a new .gif file in the supplementary material (topology-illustration.gif) that shows the learning dynamics together with the evolution of neuron topology across different learning-rate scales, clearly illustrating the neuron collapse phenomenon.
>
> #### **Q5. Relation to Edge of Stability (EOS)**
>
> We believe our results are closely related to the Edge-of-Stability phenomenon. At the EOS, training with (S)GD typically operates at an effective learning rate close to $2/K$, which in our framework corresponds to the threshold where topological simplification occurs. This connection is discussed in the caption of Figure 1, at the end of Section 4.1, and again in Section 7.
>
> #### **Q6. Can we compute or estimate K empirically**
>
> Yes. As stated in lines 169–171, the sharpness $K$ is the largest eigenvalue of the Hessian. Many off-the-shelf tools exist for estimating this value. In our experiments (e.g., Fig. 12), we compute sharpness using the hessian-eigenthings library.
>
>
> ### References
>
> [1] Gradient Descent on Neural Networks Typically Occurs at the Edge of Stability, ICLR 2021
>
> [2] Loss of plasticity in deep continual learning, Nature 2024
>
> [3] Hausdorff Dimension, Heavy Tails, and Generalization in Neural Networks, NeurIPS 2020
>
> [4] Tensor Programs I: Wide Feedforward or Recurrent Neural Networks of Any Architecture are Gaussian Processes, NeurIPS 2019
>
> [5] Mean-field theory of two-layer neural networks: dimension-free bounds and kernel limit, COLT 2019

---

### Official Review · Reviewer_s1Q7 · 2025-10-30

**Soundness:** 2
**Presentation:** 3
**Contribution:** 2
**Rating:** 4
**Confidence:** 2

**Summary:**

The paper analyses how the topology of neurons changes under learning rules (SGD, Adam). Model parameters are defined as a set of neurons, i.e. subset of parameters that are permutation-equivariant to the learning rule. First, the authors show that if the update rule is permutation-equivariant and continuous, then it provides a bi-Lipschitz mapping between the neurons at subsequent optimization steps which preserves topological properties. With the same assumptions, the main theorem establishes that the learning algorithm corresponds to a homeomorphism for sufficiently small learning rates. This implies that the evolution of neurons during training is governed by a homeomorphisms which preserve the topological properties of the manifolds. On the contrary, when the learning rate is large, the topology of neurons can change up to some degree. These results imply that there is a certain value of learning rate that governs the two phases of training dynamics. Moreover, the authors show that in the topology-preserving scenario, the learning rule corresponds to a measure preserving bijection. The work demonstrates that the proposed theory holds for common optimization methods GD and Adam. The authors verify the proposed theoretical framework with experiments with regression and MNIST classification problems.

**Strengths:**

- The paper proposes a theoretical framework characterizing the neurons’ topology under learning algorithms which does not rely on specific architectures or loss functions. The authors show that the proposed theory holds for common optimization methods such as GD and Adam.
- The experimental results demonstrate that large learning rates lead to change in topological characteristics while small learning rates preserve the topological properties as expected from the proposed theory.

**Weaknesses:**

- While theoretical results consider large and small learning rates separately, more complex learning rate schedulers are commonly used in practice. Can the proposed theoretical framework characterize the evolution of neurons' topology when the learning rate decreases and/or increases during training?
- The benefits for the practical application (for example, relation to model performance, choice of learning rate schedule, initialization choice, etc.) are not clear. How the proposed methodology can guide the training process in practice?
- The experimental part is limited. As far as I understand, the provided experiments focus only on fully connected layers and small neural networks.
- Some relevant works were not mentioned: arXiv:2401.03824v1, arXiv:2012.15834v2, arXiv:2102.00485v2

**Questions:**

Please, see weaknesses.

---

> ### Author Response · Authors · 2025-11-24
> **Response to Reviewer s1Q7**
>
> We thank the reviewer for the constructive feedbacks. Below we address each concerns raised by the reviewer.
>
> #### **W1. While theoretical results consider large and small learning rates separately, more complex learning rate schedulers are commonly used in practice. Can the proposed theoretical framework characterize the evolution of neurons' topology when the learning rate decreases and/or increases during training?**
>
> Yes. As stated at the beginning of Section 5, our theoretical framework applies to any update rule that satisfies the assumptions listed therein, which includes most commonly used learning algorithms. We intentionally use abstract terms such as "neurons" and "update rules" to make sure the results remain applicable to a broad family of optimization schemes.
>
> In particular, the theory does not require the learning rate to be constant. The key quantity governing topological evolution is the instantaneous relationship between the learning rate $\eta$ and sharpness $K$ at a certain time point $t$. As long as the update at each step satisfies the assumed regularity conditions, the topology evolves according to whether $\eta K$ stays below or crosses the critical value $1$.
>
> Therefore, a learning-rate scheduler simply introduces a time-varying $\eta_t$, and the topology evolves accordingly. If $\eta_t$ passes the critical point, the system can pass the transition point; this perspective may offer a way to interpret the behavior of specific scheduling strategies such as warm-up.
>
> #### **W2. The benefits for the practical application (for example, relation to model performance, choice of learning rate schedule, initialization choice, etc.) are not clear**
>
> Thank you for raising this important point. To provide partial empirical support, we added an additional experiment showing that loss of plasticity (LoP), a well-documented phenomenon in continual learning, can arise as a direct consequence of topology coarsening (See Appendix B, Fig. 14; content added in the rebuttal period are markded blue). This illustrates one practical implication of our framework: topology changes can restrict the model's ability to acquire new tasks, offering a principled explanation for representational rigidity observed in practice.
>
> More broadly, our theory provides a mechanism-level perspective that may inform several practical aspects:
>
> - **Model performance.** The reachable solution space is constrained by the topology induced during training. Understanding when topology is preserved or collapsed may help explain why different training configurations converge to solutions with different generalization behavior.
> - **Learning rate schedulers.** Since topological transitions occur when $\eta K$ crosses $1$, learning-rate variations can influence whether the network stays in a stable, topology-preserving regime or enters a topology-changing regime. This may offer a conceptual lens for understanding scheduling strategies such as gradual warm-up or aggressive decay.
>
> Overall, while our work does not aim to propose new training heuristics, it provides a unified theoretical framework that helps interpret several practical behaviors observed in deep learning. Please refer to the General Response for a more detailed discussion.
>
> #### **W3. The experimental part is limited. As far as I understand, the provided experiments focus only on fully connected layers and small neural networks.**
>
> We thank the reviewer for pointing this out. In response, we have added two additional experiments: a multi-layer network experiment (Figure 13) and an experiment verifying the connection between large learning rates and loss of plasticity (Figure 14). Both results are included in Appendix B. These experiments further verifies our theoretical predictions.
>
>
> #### **W4. Some relevant works were not mentioned**
>
> We thank the reviewer for suggesting the additional related work. We have included them into the revised paper. At the same time, we would like to clarify the positioning of our contribution. To the best of our knowledge, our work is the first to provide a theoretical analysis of neuron-level topology arising from learning dynamics, in a manner that applies to a broad class of optimization algorithms rather than a specific training setup. We believe this perspective is complementary to the works mentioned above.
>
> ### References
>
> [1] Loss of plasticity in deep continual learning, Nature 2024

---

### Official Review · Reviewer_WTqQ · 2025-11-01

**Soundness:** 3
**Presentation:** 3
**Contribution:** 2
**Rating:** 4
**Confidence:** 3

**Summary:**

This theoretical paper investigates the implications of permutation equivariance in learning algorithms under continuity and smoothness assumptions.
It focuses on a *neuron manifold*: a geometric object formed by representing each neuron as a particle with its input and output weights as coordinates (akin to a point cloud for a finite number of neurons, and a manifold in the infinite limit).
The core contribution of the paper is to derives conditions under which the evolution frome one timestep to the next of the neuron manifold is a surjective or homeomorphic/diffeomorphic mapping.
Furthermore, the authors identify a so-called *topological critical point* which is a specific learning rate inversely proportional to a Lipschitz-like constant and below which the neuron manifold evolves homeomorphically, preserving its topology.

This critical point can actually be envisoned as a local value as it depends on the local geometry of the loss landscape and therefore also as a dynamical value changing with the parameter during the training process.
This perspective allows for different regimes (namely: topological invariance and *topological breakdown regimes*) to occur during a single training.

**Strengths:**

The paper provides a theorem relating the learning rate and the topological nature of the update rule which describe when the topological properties of the neuron manifold is preserved.
The results on no-merging or splitting could be interesting on its own to understand redundancy in the neuron's computations.
The paper also establishes a connection between the introduced *topological critical point* with the progressive sharpening phase and edge of stability regime, which implies the topological breakdown regime.

The paper focuses on discrete updates, which is what is used in practice while theory often relies on continuous formalism.

Overall, the paper seems technically sound, with detailed proofs in the appendix, and is well presented: assumptions and definitions are clearly stated (notably in Section 3).
The main conclusion, while not so surprising, is expressed cleanly and under well-defined conditions.
The experiments should be reproducible based on information provided in the paper.

**Weaknesses:**

The scope could be a bit narrow as it is not clear whether if (or how) the framework can be used with multiple layers. The paper does not provide an explicit mapping between network parameters and the neuron point clouds in the multilayer case and experiments are also restricted to shallow architectures.

There is limited motivation for why preserving or breaking topological invariance could be useful or the motivation is not clearly justified.
Repeated claims about topological simplification, coarsening, and reduced expressivity and model capabilities in the topological breakdown regime, while theoretically valid, do not translate into actionable insights: the paper does not demonstrate such merging happen or that it affects model performance hence claims that topological simplification reduces expressivity seems overstated.
It would be convincing to see a "failure mode" that only happens in the topological breakdown regime and prevents learning on some task.
The link with grokking (line 279) is not clear and therefore seems a bit too speculative, and the application to learning rate scheduling (line 464) is not justified: why would one want to fix the topology of the neuron manifold ?

**Concern on experimental validation**: the theory predicts that under P1 and P2-K, the update $\hat U^{(t)}$ is continuous (theorem 1). Since connectedness is preserved by continuous maps, one would expect the number of connected components to remain constant.
However, Figure 4 (c) and the Betti numbers in Figure 5 show an increase in apparent connected components, which the authors interpret as evidence of topological breakdown at large learning rates.
This interpretation seems questionable: topological breakdown should involve merging (quotienting) of neurons, and if I understood correctly not creating more components (as explained the upper bound of eq 7 always holds).
My understanding is that the apparent increase in connected components could instead result from discretization artifacts: after stretching between discrete points, they can appear as disconnected clusters.
Thus, Figures 4 (c) and 5 appear inconsistent with the theory, while Figure 3 (c) and its additional intersections aligns more closely with expected behavior (merging only).

## Minor and additional feedback
- line 247: confusing sentence "once merged" should be changed to something like "whereas if two neurons are merged"
- figure 12 in appendix mentions that smaller stepsize models are trained for longer. I imagine you are reffering to the number of steps but the plots shows less epochs so it can be confusing. Moreover it is stated that small stepsize models are trained for longer to ensure convergence but the left plot does not seem to have converged yet. Finally the values of 1/K, a flatness measure (sharpness inversion) are suprising as larger stepsize end up with values an order of magnitude higher than a small stepsize (that is larger stepsize leads to sharper parameter), do you confirm these results ?
- line 355: the claim "The loss can be stably optimized only when the topology of the neurons is preserved." is unclear, are you referring to the interval $\eta \in [\frac{1}{K}, \frac{2}{K}]$ and if yes is the claim no inverted ?

**Questions:**

1. **Scope of the neuron manifold concept**
Is the notion of a neuron manifold still meaningful for MLPs with multiple hidden layers? Could each layer be viewed as an independent, partially redundant manifold, or does weight sharing across layers make this interpretation invalid?

1. **Theory–experiment alignment**
Could the authors clarify how Theorem 1 relates to the observations in Figures 3(c), 4(c), and 5, especially given the apparent inconsistency between the predicted topological continuity and the increase in connected components discussed above?

1. **Motivation**
From a practical standpoint, what is the benefit of constraining or preserving the topology of the neuron manifold? Could this have implications for interpretability or do you envision a failure mode where model expressivity could actually be significantly reduced ?

---

> ### Author Response · Authors · 2025-11-24
> **Response to Reviewer WTqQ (Part 1)**
>
> We thank the reviewer for the constructive feedback. Below we address each of the concerns raised by the reviewer.
>
> #### **W1. The scope could be a bit narrow as it is not clear whether if (or how) the framework can be used with multiple layers.**
>
> Thank you for raising this point. Our theoretical framework extends directly to neural networks of arbitrary depth. To further substantiate this, we have added a multi-layer experiment in the revised manuscript (Appendix B, Fig. 13). All additions made during the rebuttal period are highlighted in blue.
>
> Formally, in a standard feed-forward network, every pair of consecutive layers exhibits a permutation symmetry over the hidden units within that two-layer block (which we refer to as a bilayer). A bilayer can be written as
> $$g_k(x) := \sum_i v^{(k)}_i\sigma\left(\left<w_i^{(k)}, x\right>\right),$$
> where the permutation symmetry over the index $i$ is explicit. A deep network can therefore be decomposed as
> $$f = g_D \circ … \circ g_2 \circ g_1.$$
> Each $g_k$ possesses its own permutation symmetry, and our theory applies to each bilayer separately. Because the symmetry groups act on disjoint sets of neurons, the corresponding topological structures can be analyzed independently, hence the theory scales naturally to networks of any depth.
>
> **Technical note.** In practice, bilayers may partially overlap depending on architectural details. This does not affect applicability: the symmetry associated with each bilayer remains well-defined, and the topological characterization derived from our framework can be applied to each such symmetry group individually.

---

> ### Author Response · Authors · 2025-11-24
> **Response to Reviewer WTqQ (Part 2)**
>
> #### **W2. There is limited motivation for why preserving or breaking topological invariance could be useful or the motivation is not clearly justified.**
>
> Thank you for pointing out the need for clearer motivation. We agree that articulating the relevance of preserving vs. breaking topological invariance is essential. Below we summarize why the phenomenon studied in our paper is broadly meaningful for machine learning, and why its implications also extend to several related scientific domains.
>
> - **Core ML-Theoretic Motivation.** A central theme in statistical learning theory is that the generalization behavior of models is governed by the structure and size of the solution space. Our result reveals a previously unrecognized mechanism: the solutions that are reachable by gradient-based learning are strongly constrained by the topology of initialization, even when the loss landscape itself admits many equivalent optima. This suggests that topology may be an essential factor shaping generalization, implicitly biasing optimization beyond classical notions such as norm-based regularization. This fact could in turn lead to new theories for understanding the generalization behavior of neural networks, which we are currently working on as future work.
> - **Practical ML Motivation.** Empirically, seemingly minor "cosmetic" choices, such as parameterization, normalization, or architectural conventions, often have disproportionate impact on the training process. Our framework provides a principled interpretation of this phenomenon: topology-preserving modifications should leave training behavior intact, and topology-changing modifications can collapse or merge regions of the solution space, thereby inducing qualitatively different learning dynamics or enabling escape from certain undesirable configurations. This gives rise to a potential design principle: preserve topological invariants when robustness is desired, and deliberately break them when inducing new learning phases is beneficial. While detailed exploration is future work, this provides a conceptual map for interpreting and designing training heuristics.
> - **Connection to Mathematics (Topology & Geometry).** Understanding deep learning has lacked a precise mathematical language for describing symmetry-induced constraints. Our work establishes a direct correspondence between neural learning dynamics and topological structures, a link to well-developed tools in topology and differential geometry. This connection has the potential to formalize phenomena that currently rely only on empirical intuition.
> - **Relevance to Neuroscience.** An important line of thought in neuroscience is to understand our brain, the biological collection of neurons, as a manifold, whose topological and geometrical properties encode information (e.g., see[1]). It is no coincidence that artificial neural networks are used and identified as mathematical models of the brain (for example, the cerebellum is often modeled as a fully connected feedforward network [2]). Therefore, our work may be further extended to help us understand the biological brain and advance neuroscience.
> - **Relevance to Physics.** As mentioned in the main paper, topology and symmetries are fundamental objects in physics that allow us to characterize the dynamics of different matter and materials (e.g. see [3]). Under our theory, changes in network topology correspond directly to phase transitions in physics (where each different topology corresponds to a different phase). This can further help us understand various phase transition-like behaviors in deep learning, and may also help us inspire and advance theoretical physics.
>
> We hope this clarifies why topological invariance is not merely a mathematical curiosity but a phenomenon with meaningful implications across learning theory, practice, and related scientific fields. We have included some of the above points in the revised version of the manuscript.

---

> ### Author Response · Authors · 2025-11-24
> **Response to Reviewer WTqQ (Part 3)**
>
> #### **W3. The link with grokking (line 279) is not clear and therefore seems a bit too speculative, and the application to learning rate scheduling (line 464) is not justified: why would one want to fix the topology of the neuron manifold ?**
>
> Thank you for pointing this out. We agree that the statements at line 279 and line 464 describe potential future directions rather than established conclusions.
>
> **Regarding grokking.** Our intention was not to assert a direct causal link between topology changes and grokking. Instead, our result establishes a mathematically well-defined notion of a transition point at which the topology of the solution manifold changes. Because grokking and other emergent phenomena are often described as phase-transition related behaviors, our framework may offer a new lens for studying such phenomena. This is a conceptual possibility, not a claim about grokking itself.
>
> **Regarding learning rate scheduling.**  As we mentioned in the paper, when the learning dynamics preserve topology, the optimization map remains locally invertible and stable; in this regime, the network is constrained in a rather small space and the model is able to explore properties other than topological ones.
>
>
> We hope this clarifies that both discussions are framed as speculative avenues motivated by the theory, rather than definitive claims.
>
> #### **W4. Concern on experimental validation**
>
> We agree with the reviewer's interpretation: the emergence of additional connected components in the finite-width experiments is a discretization artifact. In the infinite-width limit, the manifold is continuous and its topology cannot increase.
>
> More precisely, Lemma 2 shows that when $\eta K$ becomes sufficiently large, the pairwise distance between neurons can grow exponentially over time. In the continuous (infinite-neuron) limit, this results in a stretched but still connected manifold. However, when the number of neurons is finite, the discretized samples along this manifold may become separated enough to appear as distinct connected components in the empirical graph representation.
>
> Thus, the additional components observed experimentally do not contradict the theory; they reflect a finite-sample effect arising from representing a continuous manifold with a limited number of neurons.
>
> ### Minor and additional feedback
>
> #### **line 247: confusing sentence "once merged" should be changed to something like "whereas if two neurons are merged"**
>
> Thank you for pointing this out. We agree with the reviewer, and we have revised the sentence in the manuscript accordingly.
>
>
> #### **figure 12 in appendix mentions that smaller stepsize models are trained for longer...***
>
> Thank you for catching this. There is a typo in the caption of Figure 12. The caption of the two figures should be switched. After correcting the captions, the results align with our conclusions: with a small learning rate, the value of $1/K$ decreases, and once it becomes smaller than $\eta$, the model passes the topological critical point ($\eta K > 1$), after which the topology starts to change.
>
> #### **line 355: the claim "The loss can be stably optimized only when the topology of the neurons is preserved." is unclear, are you referring to the interval and if yes is the claim no inverted ?**
>
> The statement refers to the interval $\eta \in (0, 1/K]$.
>
> ### Questions
>
> Below we answer each of the questions asked by the reviewer.
>
> #### **Q1. Scope of the neuron manifold concept.**
>
> Please see our detailed response to W1 above.
>
> #### *Q2. Theory-experiment alignment.*
>
> Please see our response to W4, where we explain why the finite-width discretization leads to the observed behavior.
>
> #### **Q3. Motivation.**
>
> Please see our response to W2 regarding the motivation and broader implications of topological invariance.
>
> ### References
>
> [1] A neural manifold view of the brain, Nature Neuroscience 2025
>
> [2] Task-dependent optimal representations for cerebellar learning, Elife 2023
>
> [3] Symmetry and Topology in Non-Hermitian Physics, Physical Review X 2024

---

> > ### Comment · Reviewer_WTqQ · 2025-11-25
> >
> > Thank you for your answers, I have a quick follow-up on the bilayer formalism, which underpins the broader applicability to multilayer networks, please tell me if I misunderstood your proposition.
> >
> > To write $f=g_D\circ\dots\circ g_2\circ g_1$, I understand that no weights are shared by parametrized functions $g_i$ and $g_{i+1}$. In that case, how do you model an even number of layers using bilayers ?
> > Moreover, while it *possible* to decompose a $4$-layer network and looking at the manifold formed by layer pairs $1-2$ and $3-4$ (as it is done on figure 13 in appendix), there is another manifold formed by the layer pair $2-3$ which seems equally valid.
> > Line 219, it is proposed to see the entirety of the neurons as a set $S\subset \mathbb R^D$. While I believe that the results for the shallow case extend to any particular bilayer since they are basically shallow networks, it is still not clear how the framework combines bilayers manifolds into a whole.
> > At which level do the generalized results apply ? At the global level (all parameters at the same time) or on each subset of bilayer parameters ?

---

> > > ### Author Response · Authors · 2025-11-25
> > >
> > > Thank you for the follow-up question. Your understanding is correct, and we are happy to clarify the points you raised.
> > >
> > > 1. **On the bilayer decomposition.** The decomposition $f = g_D \circ \cdots g_2 \circ g_1$ is meant as an illustrative way to highlight that any pair of successive layers exhibits a permutation symmetry among the neurons within that pair. It is not essential that layers have to be grouped as $(1,2), (3,4)$ etc. Any two adjacent layers $k$ and $k+1$ form a valid bilayer with its own permutation symmetry.
> > > 2. **On layers shared by two bilayers.** It is indeed possible for a layer to appear in two bilayers (e.g. layer 2 in pairs $(1,2)$ and $(2,3)$). In this case, this layer simply satisfies the theory for both sides. Understanding how these "double constraints" interact and further influence the learning dynamics is an interesting direction for future work.
> > > 3. **On Figure 13 and the 2–3 bilayer.** You are also correct that layers 2-3 form another valid bilayer in Figure 13. However, in that experiment, we restricted the output of layer 2 to be 1-dimensional so that the neuron manifold of the 1-2 bilayer can be visualized in 2D. Under this choice, the 2-3 bilayer contains only one neuron, making the case non-informative. This is why we did not plot the manifold for the 2-3 pair in that figure, though the theory still applies.
> > > 4. **On the "entirety of the neurons" (line 219).** In Sections 3 and 4, we always only focus on one specific (but arbitrary) set of neurons on which the update rule has permutation equivariance. The result should therefore be interpreted as applying to each bilayer separately.

---

### Author Response · Authors · 2025-11-24
**General Response (Part 1)**

Several reviewers have raised overlapping questions and concerns, and we would like to provide a unified discussion here to address these points collectively. We additionally outline the revisions incorporated into the manuscript.

### **Notes on the Revised Version**

In the revised manuscript, we have incorporated the suggestions from multiple reviewers by adding further discussions, references, and clarifications, as well as correcting a number of typos. We sincerely thank all reviewers for their constructive feedback, which has helped improve the clarity and completeness of the paper.

To address the concerns regarding empirical validation, which appeared in several reviews, we have added two additional experiments described in Appendix B. The first experiment provides complementary low-dimensional visualization results for multi-layer networks, confirming that our theory extends beyond two-layer models. The second experiment evaluates performance and examines topological simplification in a continual-learning setting, showing that the Loss of Plasticity (LoP) phenomenon [1] can arise as a consequence of topological simplification, thereby offering a potential explanation for LoP.

We believe that these added discussions and experiments substantially strengthen our work and help address reviewer concerns regarding the practical relevance and broader applicability of our theoretical framework. All newly added content introduced during the rebuttal period is marked in blue.

Moreover, we have included a new .gif file in the supplementary material (topology-illustration.gif) that visualizes the learning dynamics together with the evolution of neuron topology across different learning-rate scales. This animation clearly illustrates the topological breakdown predicted by our theory.

### **Discussion on the Practical Relevance**

Some reviewers expressed concerns regarding how this work may be useful in practice. In addition to the explanation of the LoP phenomenon discussed above, we believe this work also has several meaningful implications.

First of all, our work provides a mechanism-level perspective on how gradient-based learning shapes the solution space of neural networks, and we believe this has several meaningful practical implications. At the theoretical level, a central theme in statistical learning is that generalization depends on the structure and size of the reachable solution space. Our results reveal a previously unrecognized mechanism: the topology induced by initialization, and how it is preserved or simplified during training, can fundamentally constrain which solutions are reachable, even when many equivalent optima exist in the loss landscape. This suggests that topology may be an essential component of the implicit bias governing generalization, offering an alternative viewpoint to classical norm-based or margin-based analyses.

On the practical side, our framework helps explain why different training configurations can lead to different optimization behaviors. Since the topology of the reachable solution space depends on whether $\eta$ ηK remains below or crosses the critical point, training choices that influence $\eta$ (such as, learning-rate schedules) directly determine whether the dynamics stay in a stable, topology-preserving regime or enter a topology-changing regime. This provides a conceptual lens for understanding why strategies such as warm-up, cyclical schedules, or aggressive decay can dramatically alter training outcomes: topology-preserving phases maintain the structure of the solution space, while topology-changing phases can collapse or merge regions of that space, potentially enabling qualitatively different behaviors. This perspective thus offers a principled way to interpret how parameterization, normalization, and learning-rate scheduling impact model performance and generalization.



Beyond machine learning, our framework establishes a direct correspondence between neural learning dynamics and classical objects in topology and differential geometry, offering mathematical tools to formalize behaviors that are currently described only heuristically. This perspective resonates with neuroscience, where neural activity manifolds and their topological and geometrical structures are believed to encode information, and with physics, where topology and symmetry characterize different phases of matter. In this sense, changes in network topology can be viewed as phase transitions, potentially illuminating the phase-transition-like phenomena observed in deep learning and suggesting connections to theoretical physics.

---

> ### Author Response · Authors · 2025-11-24
> **General Response (Part 2)**
>
> ### **Discussion on the Finite-Neuron Setting and the Role of Topology**
>
> Several reviewers raised the concern that real neural networks contain only finitely many neurons, and that under the discrete topology, notions such as continuity and homeomorphism may appear trivial. We would like to clarify why our topological perspective remains both meaningful and nontrivial in practice.
> First of all, a common approach in theoretical deep learning is to analyze learning dynamics in the large-width (infinite-width) limit. This perspective has been widely accepted in many influential theoretical works, such as the NTK theory [2], mean-field theory [3], and the tensor program [4]. In this regime, the neuron set can be viewed as a discretization of an underlying continuous manifold, and topology becomes the natural language for describing how the learning dynamics deform or preserve this manifold. Our work follows this tradition: topology is not used as an artificial construct, but as the mathematically precise framework that emerges when neuronal populations are interpreted in the infinite-width limit. This viewpoint is not only standard in theory but also aligns with perspectives in other fields such as neuroscience, where neural populations are often modeled as low-dimensional manifolds [5].
>
> At the same time, even when one works strictly with a finite discrete set of neurons, our conclusions remain far from trivial. Under the discrete topology, continuity implies that once two neurons collapse into the same point, no subsequent gradient-based update can separate them again; homeomorphism (in the small–learning-rate regime) implies that such collapses cannot occur in the first place. These behaviors are direct consequences of the permutation symmetry inherent in neural architectures and would not hold in general. For example, consider a simple model $f(x,y) = ax + by$ (where $a \neq b$). Even when $x = y$, they can still be separated under gradient descent. The irreversibility of collapse and the impossibility of spontaneous separation are therefore substantive, nontrivial structural properties of neural networks, predicted by our theory, even in the purely discrete setting. Moreover, our LoP experiment above confirms that such neuron collapse is a practically observable phenomenon.
>
> In summary, whether one adopts the infinite-width manifold interpretation or the discrete-topology interpretation, the topological viewpoint provides meaningful, precise, and nontrivial insight into realistic learning dynamics. Far from being vacuous in finite-width networks, our results reveal structural constraints on neuron evolution that arise specifically from the permutation symmetry of neural architectures and the continuity of gradient-based updates.
>
> ### **References**
>
> [1] Loss of plasticity in deep continual learning, Nature 2024
>
>
> [2] Neural Tangent Kernel: Convergence and Generalization in Neural Networks, NIPS 2018
>
> [3] Mean-field theory of two-layer neural networks: dimension-free bounds and kernel limit, COLT 2019
>
> [4] Tensor Programs I: Wide Feedforward or Recurrent Neural Networks of Any Architecture are Gaussian Processes, NeurIPS 2019
>
>
> [5] A neural manifold view of the brain, Nature Neuroscience 2025

---

### Meta-Review · Area_Chair_dBnM · 2026-01-01

**Summary:**

* Two reviewers (Ws8X, 9kDK) argue the main theorem is close to trivial in the finite/discrete setting and that the manifold/topology preservation framing risks overclaiming. The addition of Theorem 2 partially addresses this, but it is late, not deeply validated, and arguably not yet the paper's conceptual center. For this reason, even after rebuttal, that concern remains unresolved issue because it attacks the interpretation and significance of the main result.
* Rebuttal meaningfully improves things and resolves may clarification questions by the reviewers, but doesn’t fully close the gap. The added LoP experiment + clarifications help to understand the practical relevance, and the new Theorem 2 goes in is the right direction. But it may still read as insufficiently validated / insufficiently integrated into the paper's main text to resolve the main weakness.
* Supportive reviews gave 4. WTqQ, s1Q7, 6rk3 are around borderline and still flag limited experiments and/or motivation.
* I believe the paper contains interesting ideas, but would benefit from a resubmission.

**Reviewer Concerns:**

Reviewer WTqQ:
* [Addressed] Scope beyond shallow networks clarified via a bilayer decomposition. The authors added multi-layer experiments and explicit explanation of applicability to deep networks.Practical relevance strengthened through discussion of topology as an implicit bias, learning-rate induced regime changes, and an added Loss of Plasticity experiment. Theory-experiment mismatch explained as a finite-width discretization artifact. Speculative claims (grokking, LR scheduling) are left  as future work.

Reviewer s1Q7:
* [Addressed] Practical implications clarified, with new LoP experiment linking topology coarsening to continual-learning failure. Limited experiments partially mitigated by adding multi-layer and continual-learning experiments. Missing related work added.

Reviewer 6rk3:
* [Addressed] Artificial manifold initialization justified, clarified relevance to random initialization and post-training low-dimensional structure. Applicability to CNNs and Transformers explained, including feed-forward and attention blocks. Discrete vs. infinite-width topology clarified. Nontriviality of collapse and irreversibility emphasized. Relation to Edge of Stability, sharpness estimation, and prior topology work clarified.

Reviewer Ws8X:
* [Addressed] Claim that topology coarsening reduces expressivity supported by theoretical arguments and new LoP experiment. Clarified that stability refers to continuity/injectivity. Use of TDA justified for experiments.
* [Partially addressed] Finite-width concern responded to with dual viewpoints and LoP experiment, but reviewer skepticism likely remains.

Reviewer 9kDK:
* [Partially addressed] Discrete-topology critique acknowledged. Authors clarified limits of strict homeomorphism interpretation and reframed claims to avoid overstatement.
* [Addressed] Added Theorem 2 introducing a neighborhood-based topology where collapses are non-measure-zero, strengthening practical relevance. Infinite-width/manifold interpretation further justified. LoP experiment cited as evidence of real neuron merging. Layerwise nature of permutation symmetry clarified and accepted as intended scope.

In general, the rebuttal is strong and solid.

**Reviewer Scores:**

* Reviewer WTqQ: Would have kept the score → 4 (from 4).
* Reviewer s1Q7: Would have kept the score --> 4 (from 4).
* Reviewer 6rk3: Would have kept the score --> 4 (from 4) They still viewed experiments and generalization relevance as limited. Rebuttal answers questions well, but probably not enough to raise the score.
* Reviewer Ws8X: May have slightly increased --> 2 or 4 (from 2) The rebuttal addresses core concerns, but their confidence was high and skepticism fundamental.
* Reviewer 9kDK: May have slightly increased --> 2 or 4 (from 2). This reviewer explicitly acknowledged improvements, extra experiments, and engaged deeply. The addition of Theorem 2 directly targets their main critique.

---

### Decision · Program_Chairs · 2026-01-26

Reject